# THE PRICE OF ROBUSTNESS: STABLE CLASSIFIERS NEED OVERPARAMETERIZATION

**Jonas von Berg & Adalbert Fono & Massimiliano Datres & Sohir Maskey**[*]**& Gitta Kutyniok**[†]
Ludwig-Maximilians-Universität München
Munich, Germany
Munich Center for Machine Learning (MCML)
`{berg,fono,datres,kutyniok}@math.lmu.de`
`sohir.maskey@aleph-alpha-research.com`

## ABSTRACT

The relationship between overparameterization, stability, and generalization remains incompletely understood in the setting of discontinuous classifiers. We address this gap by establishing a generalization bound for finite function classes that improves inversely with *class stability*, defined as the expected distance to the decision boundary in the input domain (margin). Interpreting class stability as a quantifiable notion of robustness, we derive as a corollary a *law of robustness for classification* that extends the results of Bubeck and Sellke beyond smoothness assumptions to discontinuous functions. In particular, any interpolating model with $p \approx n$ parameters on $n$ data points must be *unstable*, implying that substantial overparameterization is necessary to achieve high stability. We obtain analogous results for (parameterized) infinite function classes by analyzing a stronger robustness measure derived from the margin in the codomain, which we refer to as the *normalized co-stability*. Experiments support our theory: stability increases with model size and correlates with test performance, while traditional norm-based measures remain largely uninformative.

## 1 INTRODUCTION

The generalization behavior of overparameterized neural networks presents fundamental challenges to classical statistical learning theory. Traditional complexity measures, such as parameter counts or spectral norms of weights, form the basis of many generalization bounds, including those derived from VC dimension theory (Sain, 1996) and Rademacher complexity (Bartlett & Mendelson, 2002). However, these approaches do not adequately explain several empirical phenomena, e.g., *double descent* (Belkin et al., 2019) and *benign overfitting* (Bartlett et al., 2020). The occurrence of double descent illustrates that the test error, after initially increasing near the interpolation threshold, can improve as the model size continues to grow. Similarly, the phenomenon of benign overfitting demonstrates that models that perfectly interpolate noisy training data can nonetheless achieve strong generalization. Such findings expose the limitations of norm- and size-based complexity measures as predictors of generalization.

A large-scale empirical study evaluating more than forty complexity measures found that many norm-based quantities not only fail to correlate with generalization, but often even correlate negatively (Jiang et al., 2019). Beyond optimization-related metrics, one of the few quantities that consistently correlated with generalization was the margin, i.e., the distance to the decision boundary, closely related to the notion of (co-)stability we develop in this work. This aligns with an emerging perspective: generalization in modern networks is governed less by model size or norms, and more by the *stability / robustness* of predictions under input perturbations (Soloff et al., 2025; Ghosh & Belkin, 2023; Zhang et al., 2022). Related insights also arise from the literature on algorithmic stability (Bousquet & Elisseeff, 2002) and flat minima (Keskar et al., 2017). However, most theoretical results in this direction are restricted to linear models.

---

[*]now at Aleph Alpha Research
[†]University of Tromso, DLR-German Aerospace Center

An exception is the *universal law of robustness* of Bubeck & Sellke (2021), which, under mild distributional assumptions, establishes a formal link between robustness, generalization, and over-parameterization: smoothness and overparameterization need to balance in order to ensure good generalization while overfitting. The *law of robustness* relies on the assumption that the function class is Lipschitz, which makes it inadequate for classifiers whose codomain is discrete by design. We therefore take a step toward the open challenge posed in Bubeck & Sellke (2021, p. 4): "[...] it is an interesting challenge to understand for which notions of smoothness there is a tradeoff with size." Specifically, we introduce *class stability* and *normalized co-stability* as geometric smoothness measures that extend robustness laws to classification. In fact, replacing Lipschitz continuity is essential: simply focusing on the Lipschitz constant of an underlying score function $g$, where the classifier is of type $f := \arg\max \circ g$, is not informative. In particular, since $g$ can be arbitrarily rescaled without changing the predictions of $f$, its Lipschitz constant does not need to reflect the geometry of the decision boundary (Liu & Hansen, 2024).

**Paper Roadmap.** We discuss related work in Section 2. Section 3 introduces class stability and the isoperimetry assumption, a concentration property of the data that underlies our analysis. Section 4 presents a generalization bound for finite hypothesis classes and examines its implications for overparameterization. In Section 5, we extend the framework to infinite function classes via the notion of normalized co-stability. Our theoretical predictions are tested experimentally on MNIST and CIFAR-10 in Section 6. Finally, Section 7 concludes with a discussion of open directions.

**Contributions.** We provide a summary of our main results.

1) We prove that, under an isoperimetry assumption on the data distribution, the data-dependent Rademacher complexity of a finite hypothesis class of classifiers can be bounded in terms of the minimum *class stability*. This yields an improved generalization bound for discontinuous classifiers (Theorem 4), which tightens as stability increases.

2) We show that in the classically parameterized regime (#parameters $\approx$ #samples), any interpolating classifier must be unstable (Corollary 6) with high probability. Consequently, achieving both near-perfect fitting and high class stability requires substantial overparameterization of order $p \approx nd$.

3) We extend the framework to infinite function classes by considering classifiers of the form $f(x) := \arg\max \circ g_w(x)$, where $g_w$ is a parameterized Lipschitz-continuous (in both $x$ and $w$) score function. This enables us to define a robustness measure – the *normalized co-stability* –, based on output score margins, and derive a corresponding generalization bound (Theorem 13). The added regularity also results in a law of robustness for infinite function classes (Corollary 15).

4) We empirically validate our theoretical predictions on MNIST and CIFAR-10. Stability and normalized co-stability increase with network width and exhibit the same qualitative scaling as test accuracy, highlighting the central role of (normalized co-)stability in overparameterized generalization.

Taken together, our results extend the law of robustness to discontinuous classifiers and highlight stability as a central factor in understanding generalization in modern networks.

## 2 RELATED WORK

**Smoothness-based generalization.** Our work is inspired by the *law of robustness* of Bubeck & Sellke (2021), which shows that regression with Lipschitz predictors generalizes when smoothness and overparameterization are properly balanced. Subsequent works have extended this perspective: for example, Zhu et al. (2023) investigate how width, depth, and initialization affect robustness, while more recent studies Das et al. (2025) establish refined smoothness–generalization trade-offs for a wider range of loss landscapes.

**Margin-based generalization.** Classical generalization bounds combine a margin term, defined with respect to a score function, with a capacity measure – for example, spectrally-normalized margin bounds (Bartlett et al., 2017) or path-norm bounds (Neyshabur et al., 2018). Recent extensions include multi-class margin bounds in terms of margin-normalized geometric complexity (Munn et al., 2024). These approaches are closely aligned with our normalized co-stability perspective:

both control a codomain margin while coupling it to a regularity property of the score function, and both recover inverse-margin scaling.

Input-space margin bounds have also been studied, yielding that generalization is controlled by the minimum robustness radius (Sokolic et al., 2017), while sample-complexity lower bounds show that adversarial robustness increases the VC dimension (Gao et al., 2019). Our notion of *class stability* differs: it is the *expected input margin* – the average distance to the decision boundary under the data distribution – rather than a minimum or an empirical quantile. This measure is closely tied to robustness (Fawzi et al., 2016; Gilmer et al., 2018) and induces data-dependent bounds that track generalization.

**Limits of uniform generalization bounds.**  Uniform convergence–based bounds are often vacuous in overparameterized networks (Nagarajan & Kolter, 2021), since SGD appears to find solutions at a macroscopic level (supporting generalization) but with microscopic fluctuations that break uniform analyses. Our bounds remain uniform but depend on macroscopic, distribution-dependent quantities: the Rademacher complexity – our applied technique to derive generalization bounds – is controlled by stability (or co-stability). Whether this structure avoids the vacuity identified by Nagarajan & Kolter (2021) remains open.

**Stability, robustness, and implicit bias.**  Algorithmic stability (Bousquet & Elisseeff, 2002) and the flat minima literature (Keskar et al., 2017) argue that robustness under perturbations drives generalization. More recently, Zou et al. (2024) derive out-of-distribution generalization bounds based on the sharpness of the learned minima. Our contribution is to extend a stability-based perspective to discontinuous neural classifiers, both theoretically and empirically. Complementary work on implicit bias shows that gradient descent favors solutions with a small number of connected decision regions, a proxy for large input-space margin (Li et al., 2025). This suggests that optimization dynamics may implicitly favor the same geometric simplicity that our stability-based bounds capture.

**Out-of-Distribution Generalization.**  Classically and also in our analysis, generalization is based on the independently and identically distributed assumption on the data, in particular, the test data are generated from the same distribution as the training data coined In-Distribution (ID) generalization. In contrast, Out-of-Distribution (OOD) generalization aims to study the generalization performance under distributional shifts. To make the problem tractable the potential shifts are constrained to, for instance, spurious correlations or covariate shifts. In the OOD setting the connection between over-parameterization and generalization has been studied in a series of theoretical works with positive Hao et al. (2024) and negative results Sagawa et al. (2020); Wald et al. (2024).

Adversarial robustness can be viewed as a special case of OOD generalization, where the distributional shift is constrained to lie within a perturbation set Sinha et al. (2020). In this sense, our stability-based analysis is conceptually connected to OOD generalization. However, our results do not provide explicit bounds on OOD error; instead, we focus on ID generalization under the assumption that the classifier satisfies a given level of adversarial robustness expressed as the margin.

## 3    PRELIMINARIES AND NOTATION

In the following, we provide background on the key concepts underlying our analysis, namely stability, generalization, and isoperimetry. For clarity of exposition, we present our results in the binary classification setting. The extension to multi-class problems follows by a one-vs-all reduction; see Appendix F for details. Thus, let $(\mathcal{X} \times \{-1, 1\}, \mu)$ be a probability measure space with $\mathcal{X} \subset \mathbb{R}^d$ bounded and $\mathcal{F} \subset \{f \mid f : \mathcal{X} \to \{-1, 1\}\}$ a set of classifiers. The goal is to find a stable function $f \in \mathcal{F}$ minimizing a bounded loss function $\ell : \{-1, 1\}^2 \to \mathbb{R}_+$ on $n$ i.i.d. samples $(x_i, y_i) \sim \mu$. A natural loss in the classification setting is the 0–1 loss $\ell_{0\text{-}1}(y, y') := \mathbb{1}_{y \neq y'}$. In this setup, following a similar approach as in Liu & Hansen (2024), we define the *class stability* of $f$ as the expected distance of a sample to the decision boundary in $\mathcal{X}$, thereby capturing the average robustness of a classifier $f$ to input perturbations.

**Definition 1** (Margin and Class Stability). *Let $f : \mathcal{X} \to \{-1, 1\}$. The* signed distance function $d_f$ *of $f$ at $x \in \mathcal{X}$ is defined as*

$$d_f(x) := \begin{cases} d(x, f^{-1}(\{-1\})), & \text{if } f(x) = 1, \\ -d(x, f^{-1}(\{1\})), & \text{if } f(x) = -1, \end{cases}$$

*where $d(x, A) := \inf_{y \in A} \|x - y\|_2$. We define the (unsigned)* margin $h_f$ *at $x$ as the absolute value of the signed distance function,*

$$h_f(x) := |d_f(x)| = \inf\{\|x - z\|_2 : f(z) \neq f(x),\ z \in \mathcal{X}\}.$$

*The* class stability $S(f)$ *of $f$ is its expected margin under the data distribution:*

$$S(f) := \mathbb{E}[h_f].$$

**Remark 2.** *The signed distance function $d_f$ is 1-Lipschitz if $\mathcal{X}$ is path-connected. Moreover, if $\operatorname{sgn}(0) = 1$ and $f^{-1}(\{1\})$ is closed in $\mathcal{X}$, then $f$ admits the representation $f = \operatorname{sgn} \circ d_f$ (see Appendix B for details).*

Our goal is to relate the class stability to the Rademacher complexity of a function class, which, in turn, connects to *generalization* bounds through classical results (Bartlett & Mendelson, 2002). In particular, for a bounded loss $|\ell| \leq a$, the difference between the *population risk* $R_\ell(f) := \mathbb{E}[\ell(f(x), y)]$ and the *empirical risk* $\hat{R}_\ell(f) := \frac{1}{n} \sum_{i=1}^n \ell(f(x_i), y_i)$ is bounded with probability at least $1 - \delta$ over the samples by

$$\sup_{f \in \mathcal{F}} \big( R_\ell(f) - \hat{R}_\ell(f) \big) \leq 2\mathcal{R}_{n,\mu}(\ell \circ \mathcal{F}) + a\sqrt{\frac{2\log(2/\delta)}{n}}, \tag{1}$$

where $\mathcal{R}_{n,\mu}(\mathcal{G})$ denotes the *Rademacher complexity* of a general function class $\mathcal{G}$, defined as

$$\mathcal{R}_{n,\mu}(\mathcal{G}) = \frac{1}{n} \mathbb{E}^{\sigma_i, x_i} \left[ \sup_{g \in \mathcal{G}} \left| \sum_{i=1}^n \sigma_i g(x_i) \right| \right],$$

with $(\sigma_i)_{i=1}^n$ i.i.d. Rademacher random variables. To obtain a bound in Equation 1 in terms of $\mathcal{R}_{n,\mu}(\mathcal{F})$, note that $\mathcal{R}_{n,\mu}(\ell \circ \mathcal{F}) \leq C\mathcal{R}_{n,\mu}(\mathcal{F})$ holds under certain conditions on the loss, we have

$$\mathcal{R}_{n,\mu}(\ell_{0\text{-}1} \circ \mathcal{F}) \leq \frac{1}{2}\mathcal{R}_{n,\mu}(\mathcal{F}), \quad \text{i.e., } C = \frac{1}{2}, \tag{2}$$

whereas for $L$-Lipschitz losses $C = L$ holds, see Bartlett & Mendelson (2002); Shalev-Shwartz & Ben-David (2014) for detailed explanations. Overall, it therefore suffices to bound $\mathcal{R}_{n,\mu}(\mathcal{F})$ in terms of the class stability of functions $f \in \mathcal{F}$ in order to link generalization to stability. In other words, the key step is to control how well stable functions can fit random labels, which requires structural assumptions on the input distribution. We discuss in detail in Appendix A why such assumptions are unavoidable. A natural and widely used condition is *isoperimetry*, which guarantees sharp concentration for bounded Lipschitz-continuous functions (Bubeck & Sellke, 2021).

**Definition 3** (Isoperimetry). *A probability measure $\mu$ on $\mathcal{X} \subset \mathbb{R}^d$ satisfies $c$-isoperimetry if for any bounded $L$-Lipschitz function $f : \mathcal{X} \to \mathbb{R}$, and any $t \geq 0$,*

$$\mathbb{P}(|f(x) - \mathbb{E}[f]| \geq t) \leq 2e^{-\frac{dt^2}{2cL^2}}. \tag{3}$$

Isoperimetry is, for instance, satisfied by Gaussian measures and the (normalized) volume measure on Riemannian manifolds with positive curvature, such as the uniform measure on the sphere (Vershynin, 2018; Bubeck & Sellke, 2021). Consequently, under the manifold hypothesis, the relevant dimension in our bounds can be interpreted as the intrinsic manifold dimension rather than the ambient dimension.

## 4 A LAW OF ROBUSTNESS FOR CLASSIFICATION

In this section, we establish a *law of robustness for classification*, extending stability-generalization trade-offs to discontinuous functions. Classical results for smooth functions characterize robustness via the Lipschitz constant, which is ill-defined for classifiers with discrete outputs. To address this, we follow the general strategy of Bubeck & Sellke (2021) (see Appendix A for details), but replace their use of Lipschitz continuity with our notion of *class stability* (Definition 1). Formally, we proceed under the following assumptions:

(H1) $(\mathcal{X} \times \{-1, 1\}, \mu)$ is a probability space with bounded sample space $\mathcal{X}$ and c-isoperimetric[1] marginal distribution $\mu_{\mathcal{X}}$;

(H2) the considered hypothesis class $\mathcal{F}$ of classifiers $f : \mathcal{X} \to \{-1, 1\}$ is finite, that is $|\mathcal{F}| < \infty$.

These conditions ensure concentration of measure in the input space and allow complexity control via a union bound. With this structure in place, class stability can be related to the Rademacher complexity, leading to the bound stated below.

**Theorem 4** (Rademacher Bound). *Suppose Assumptions (H1) and (H2) hold, and that* $\min_{f \in \mathcal{F}} S(f) > S > 0$ *with* $\log |\mathcal{F}| \geq n$.

*1. The Rademacher complexity satisfies*

$$\mathcal{R}_{n,\mu}(\mathcal{F}) \ \leq \ K_1 \max \left\{ \frac{1}{\sqrt{n}}, \ \frac{\sqrt{c}}{S} \cdot \frac{\log |\mathcal{F}|}{n\sqrt{d}} \right\}, \tag{4}$$

*for an absolute constant* $K_1 > 0$.

*2. If, in addition,* $f^{-1}(\{1\})$ *is closed and* $\mathcal{X}$ *path-connected, the bound sharpens to*

$$\mathcal{R}_{n,\mu}(\mathcal{F}) \ \leq \ K_2 \max \left\{ \frac{1}{\sqrt{n}}, \ \frac{\sqrt{c}}{S} \sqrt{\frac{\log |\mathcal{F}|}{nd}}, \ 2 \exp\left( -\frac{dS^2}{8c} \right) \right\}, \tag{5}$$

*for another absolute constant* $K_2 > 0$.

*Proof sketch.* Equation 4 is obtained via a Lipschitz surrogate argument combined with isoperimetry. The refined bound in Equation 5 further leverages the representation $f = \mathrm{sgn} \circ d_f$ (Remark 2), using that large stability ensures $d_f$ remains well separated from the discontinuity at $0$. Complete details are provided in Appendix C. $\square$

**Remark 5.** *In contrast to Bubeck & Sellke (2021), where stability is measured by the minimal Lipschitz constant of the function class, our initial bound in Theorem 4 incurred an additional factor* $\sqrt{\log |\mathcal{F}|/n}$ *in the regime* $\log |\mathcal{F}| \geq n$. *By assuming mild regularity conditions, we can eliminate this gap and recover the same scaling as in Bubeck & Sellke (2021).*

The key insight of Theorem 4, combined with the classical generalization bound in Equation 1, is that *good generalization* can still be achieved in the highly *overparameterized* regime – provided the classifiers exhibit sufficiently *high class stability*. Indeed, the presence of $\frac{1}{S}$ in front of $\sqrt{\log |\mathcal{F}|}$ in Equation 4 and Equation 5 indicates that class stability affects the effective complexity of the model class, potentially mitigating the risks of overfitting in large models. Note that, using a uniform discretization, a finite approximation of an infinite function class parameterized with $p$ parameters over a bounded subset of $\mathbb{R}^p$ satisfies $\log |\mathcal{F}| \in \mathcal{O}(p)$. In this sense, $\log |\mathcal{F}|$ reflects the number of model parameters. Therefore, when the number of parameters $p \approx \log |\mathcal{F}|$ is much larger than $n$, the second term in the maximum in Equation 5 dominate, and the bounds becomes small if $S$ scales at least in the order of $\sqrt{\frac{p}{nd}}$.

We are now ready to state our *law of robustness for discontinuous functions*, obtained as a direct corollary of the refined Rademacher bound in Equation 5 of Theorem 4.

**Corollary 6** (Law of Robustness for Discontinuous Functions). *Assume (H1), (H2), and the additional conditions in 2. of Theorem 4 hold. Let* $p := \log |\mathcal{F}| \geq n$. *Fix* $\varepsilon, \delta \in (0, 1)$ *and consider the 0–1 loss* $\ell_{0-1}$. *There exists an absolute constant* $K > 0$ *such that, if*

*1. the minimal risk* $R^* := \min_{f \in \mathcal{F}} R_{0-1}(f)$ *satisfies* $R^* \geq \varepsilon$, *and*

*2. the sample size* $n$ *is large enough to ensure (i)* $\frac{K}{\sqrt{n}} < \frac{\varepsilon}{3}$ *and (ii)* $\sqrt{\frac{2\log(2/\delta)}{n}} < \frac{\varepsilon}{2}$,

*then with probability at least* $1 - \delta$ *(over the sample), the following holds uniformly for all* $f \in \mathcal{F}$:

$$\hat{R}_{0-1}(f) \leq R^* - \varepsilon \quad \Longrightarrow \quad S(f) < \max \left\{ \frac{3K}{\varepsilon} \sqrt{\frac{c \log |\mathcal{F}|}{nd}}, \ \sqrt{\frac{8c}{d} \log\left( \frac{6K}{\varepsilon} \right)} \right\}. \tag{6}$$

---

[1]It is worth noting that our framework can be readily extended to mixtures of c-isoperimetric distributions.

*Proof sketch.* Apply the Rademacher bound (Theorem 4) to the high-stability subset $\mathcal{F}_{S_*} := \{f \in \mathcal{F} : S(f) \geq S_*\}$. For $S_*$ chosen large enough, such functions cannot achieve empirical risk below $R^* - \varepsilon$, so any interpolating classifier with risk $\leq R^* - \varepsilon$ must lie outside $\mathcal{F}_{S_*}$, i.e., must satisfy $S(f) < S_*$. The full proof is provided in Appendix D. □

**Remark 7.** *As in Bubeck & Sellke (2021), we assume $R^\star > 0$, corresponding to a setting in which the hypothesis class does not achieve zero classification error. This accounts for inherent label noise or model misspecification. Unlike Bubeck & Sellke (2021), which assumes Lipschitz-continuous losses, our analysis directly addresses the discontinuous 0–1 loss, making it more natural for classification tasks. The overall proof strategy, however, extends to arbitrary losses provided one can derive an appropriate bound on the Rademacher complexity of the composed function class, as in Equation 2.*

**Remark 8.** *Importantly, this result also covers intrinsically discontinuous classifiers, such as quantized neural networks and spiking neural networks. Moreover, since self-attention is in general not Lipschitz-continuous Kim et al. (2021), our framework appears particularly well-suited to the analysis of overparameterization of transformers, which underlie most state-of-the-art language models.*

From Equation 6 we conclude that achieving both low training error and high stability requires parameterization on the order $p \approx nd$. This necessity arises in the high-dimensional regime, since when $d$ is large the first term in the maximum dominates for $p \geq n$. This reinforces our central message: *overparameterization may not harm generalization, but on the contrary, is necessary for achieving robustness and good fitting in classification.* Notably, modern neural networks, including large language models (LLMs) (Brown et al., 2020), are trained in heavily overparameterized regimes: Even though recent scaling laws Hoffmann et al. (2022) suggest a balance between model and data size, contemporary language models still operate deep in the overparameterized interpolation regime, where their capacity is sufficient to achieve near-zero training loss. Therefore, our result may help to understand why such models still do generalize effectively.

## 5 A LAW OF ROBUSTNESS FOR INFINITE FUNCTION CLASSES

In Theorem 4, our analysis does not straightforwardly extend to infinite function classes. The usual proof strategy via a covering-number argument requires closeness in parameter space to imply closeness in function space. In Bubeck & Sellke (2021), this is enforced via Lipschitz continuity in the parameters of the function class, but such a condition is in general meaningless for discontinuous classifiers.

To overcome this, we restrict our attention to function classes with additional structure and introduce a strengthened stability notion. Specifically, we impose a representation analogous to Remark 2, namely,

(H3) The hypothesis class has the form $\mathcal{F} = \mathrm{sgn} \circ \mathcal{G}$, where $\mathcal{G} = \{g_w : \mathcal{X} \to [-1, 1] : w \in \mathcal{W}\}$ is a parameterized family of Lipschitz functions. The parameter space $\mathcal{W} \subset \mathbb{R}^p$ is bounded with $\mathrm{diam}(\mathcal{W}) \leq W$, and the parameterization is Lipschitz:

$$\|g_{w_1} - g_{w_2}\|_\infty \leq J \|w_1 - w_2\|.$$

The extension from finite to infinite classes requires not only (i) Lipschitz continuity in $w$, but also (ii) that the scores $g_w(x)$ stay quantitatively away from zero, so that small parameter perturbations cannot cause arbitrary label flips. Class stability alone does not suffice to ensure (ii), as the following example demonstrates.

**Example 9** (Class stability does not prevent discontinuity). *Let $\mathcal{G} = \{g_w(x) = w \tanh(x) : w \in [-1, 1]\}$. The parameterization is Lipschitz since*

$$\|g_{w_1} - g_{w_2}\| \leq \|w_1 - w_2\|.$$

*For $w_1 = \frac{\varepsilon}{2}$ and $w_2 = -w_1$, $\|w_1 - w_2\| \leq \varepsilon$, yet*

$$\|\mathrm{sgn}(g_{w_1}(x)) - \mathrm{sgn}(g_{w_2}(x))\| = 2$$

*for almost all $x$. Each classifier has a single boundary point (hence high class stability), but parameter proximity does not imply classifier proximity.*

To guarantee property (ii), we introduce a new robustness measure in the codomain.

**Definition 10** (Co-margin and Co-stability). *Let $f = \text{sgn} \circ g : \mathcal{X} \to \{-1, 1\}$. The* co-margin *at $x$ is*

$$h_g^*(x) := |g(x)|,$$

*and we denote the* normalized co-margin *as*

$$\bar{h}_g^*(x) := \frac{|g(x)|}{L(g)},$$

*where $L(g)$ is the Lipschitz constant of $g$. The* co-stability *is then the expected co-margin*

$$S^*(g) := \mathbb{E}[h_g^*(x)],$$

*and the* normalized co-stability *is accordingly defined as the expected normalized co-margin*

$$\bar{S}^*(g) := \mathbb{E}[\bar{h}_g^*(x)].$$

**Remark 11** (Representation dependence). *Unlike class stability $S(f)$, which depends only on the decision boundary of $f$, the co-stability $S^*(g)$ and its normalized form $\bar{S}^*(g)$ depend on the particular representation $f = \text{sgn} \circ g$. Different score functions $g$ inducing the same classifier $f$ can yield different values of $S^*(g)$ and $\bar{S}^*(g)$. For the specific representation $f = \text{sgn} \circ d_f$ from Lemma 18, however, both measures coincide: $S^*(g) = \bar{S}^*(g) = S(f)$.*

Imposing $S^*(g) \geq S^* > 0$ ensures that scores remain, on average, a non-trivial distance away from zero. Together with (H3), co-stability provides the continuity and separation properties required for infinite-class generalization bounds.

Before turning to the formal statement of this fact, we want to discuss the relation of class stability and co-stability. The connection between input- and codomain-based margins is immediate since

$$h_g(x) \geq \frac{h_g^*(x)}{L(g)} = \bar{h}_g^*(x).$$

By $L(g)$-Lipschitz continuity, moving $x$ by $r$ changes $g(x)$ by at most $L(g)r$, so flipping the prediction requires $r \geq |g(x)|/L(g)$. Taking expectations yields

$$S(f) \geq \bar{S}^*(g). \tag{7}$$

Thus normalized co-stability lower-bounds class stability. This inequality highlights two levers for improving generalization: increasing $S^*(g)$ or decreasing $L(g)$. Importantly, $\bar{S}^*(g)$, like $S(f)$, is invariant to input rescaling and therefore serves as a natural robustness measure.

**Remark 12.** *A related ratio, $\frac{\gamma}{\mathcal{R}_f}$, appears in Bartlett et al. (2017), where $\gamma$ is the minimum margin and $\mathcal{R}_f$ a spectral complexity term controlling Lipschitzness. Empirically, Lipschitz margin training, which enforces*

$$\bar{S}^*(g) \geq c,$$

*improves adversarial robustness (Tsuzuku et al., 2018). Moreover, Béthune et al. (2022, Corollary 2) show that among maximally accurate classifiers, there exists a 1-Lipschitz solution that achieves maximal co-margins and satisfies $S(f) = S^*(g)$. In particular, the Bayes classifier admits the representation $b = \text{sgn} \circ d_b$, which fulfills these properties.*

Combining Theorem 4 with Equation 7, the Rademacher complexity of a finite function class $\mathcal{F} = \text{sgn} \circ \mathcal{G}$ can be bounded in terms of normalized co-stability as

$$\mathcal{R}_{n,\mu}(\mathcal{F}) \leq K_2 \max\left\{ \frac{1}{\sqrt{n}}, \ \sqrt{c} \frac{L}{S^*} \sqrt{\frac{\log |\mathcal{F}|}{nd}}, \ 2\exp\left(-\frac{dS^{*2}}{L^2 8c}\right) \right\},$$

where $S^* > 0$ and $L > 0$ are bounds on the minimal co-stability and maximal Lipschitz constant, respectively. Under condition (H3), the statement can be extended to infinite function classes.

**Theorem 13.** *Suppose (H1) and (H3) hold, and that $S^*(g) > S^* > 0$ and $L(g) \leq L$ for all $g \in \mathcal{G}$. Assume further that $p \geq n$. Then, for any covering precision $\tilde{\varepsilon} > 0$,*

$$\mathcal{R}_{n,\mu}(\mathcal{F}) \leq K \max\left\{ \sqrt{\frac{1}{n}}, \ \frac{L}{S^*} \sqrt{\frac{p}{nd}} \sqrt{c \log\left(1 + 60WJ\tilde{\varepsilon}^{-1}\right)}, \ 2\exp\left(-\frac{dS^{*2}}{8cL^2}\right), \ \frac{J}{S^*}\tilde{\varepsilon} \right\}, \tag{8}$$

*where $K > 0$ is an absolute constant independent of $p, n, d, S^*, c, L, J, \tilde{\varepsilon}, W$.*

*Proof sketch.* The proof follows the previously mentioned $\varepsilon$-net approach, standard in infinite-class settings. The Lipschitz continuity in $w$ (from (H3)) controls the covering number of $\mathcal{G}$ at scale $\tilde{\varepsilon}$, while co-stability ensures that small perturbations in $w$ do not induce flips through the sgn mapping. The additional term $\frac{J}{S^*}\tilde{\varepsilon}$ reflects the residual error introduced by the discretization. See Appendix E for more details. $\qquad\square$

**Remark 14.** *The factor $\frac{L}{S^*}$ shows that generalization depends jointly on the average prediction confidence $S^*(g)$ and the Lipschitz constant $L(g)$, the latter quantifying robustness of predicted probabilities. This aligns with empirical findings (Khromov & Singh, 2024; Gamba et al., 2025; Gouk et al., 2020; Sanyal et al., 2020; Béthune et al., 2022), which report that smaller Lipschitz constants typically improve generalization, and in some cases exhibit a double-descent behavior.*

We obtain with the same reasoning as in Corollary 6 the following law of robustness for Lipschitz-regular infinite function classes.

**Corollary 15** (Law of Robustness for Infinite Function Classes). *Assume (H1) and (H3), and fix $\varepsilon, \delta \in (0,1)$. Consider the 0-1 loss $\ell_{0-1}$. There exists an absolute constant $K > 0$ such that, if*

1. *the minimal risk $R^* := \inf_{f \in \mathcal{F}} R_{0-1}(f)$ satisfies $R^* \geq \varepsilon$, and*

2. *the sample size $n$ is large enough so that (i) $\frac{K}{\sqrt{n}} < \frac{\varepsilon}{3}$ and (ii) $\sqrt{\frac{2\log(2/\delta)}{n}} < \frac{\varepsilon}{2}$,*

*then with probability at least $1 - \delta$, for all $\tilde{\varepsilon} > 0$, the following holds uniformly for all $g \in \mathcal{G}$ and $f_g = sgn \circ g$:*

$$\hat{R}_{0-1}(f_g) \leq R^* - \varepsilon \quad \Longrightarrow \quad \frac{S^*(g)}{L(g)} < \max\left\{ \frac{3K}{\varepsilon}\sqrt{\frac{p}{nd}}\sqrt{c\log(1 + 60WJ\tilde{\varepsilon}^{-1})}, \ \sqrt{\frac{8c}{d}\log\left(\frac{6K}{\varepsilon}\right)} \right\}.$$

**Remark 16.** *As in Bubeck & Sellke (2021), we require $W$ and $J$ to be at most polynomial in $(n, d, p)$ so that they do not affect the asymptotic scaling. In the case of feedforward neural networks, Bubeck & Sellke (2021) further show that when the data distribution is concentrated in a ball of radius $R$, it suffices to assume that $W$ is polynomially bounded.*

Analogous to the finite-class case, we conclude that Lipschitz-regular classifiers must be overparameterized of order $nd$ to achieve both low training error and high normalized co-stability. Without sufficient parameter capacity relative to sample size and ambient dimension, robustness cannot be guaranteed: models may fit the training data, but will necessarily exhibit either large Lipschitz constants of the score function or low co-stability, reflecting weak confidence in their predictions. Thus, overparameterization emerges as a necessary condition for robustness, reflecting a structural constraint imposed by geometry and probability rather than a mere byproduct of contemporary training practice.

## 6 EXPERIMENTS

We empirically investigate how class stability and co-stability scale with model size in interpolating networks.

**Experimental setup.** We train fully connected MLPs with four or eight hidden layers and widths $w \in \{128, 256, 512, 1024, 2048\}$ on MNIST and CIFAR-10. On CIFAR-10, we additionally evaluate CNNs with widths $w \in \{128, 256, 512, 1024\}$. On MNIST, we consider Heaviside-activation MLPs to probe whether the observed scaling extends to discontinuous score functions. All models are trained to at least 99% training accuracy, yielding near-interpolating solutions.

**Class Stability.** We estimate empirical class stability $S(f)$ via adversarial perturbations. For each input, we increase the perturbation radius $r$ along a predefined grid $\mathbf{r} = (r_1, \ldots, r_n)$ until the classifier's prediction changes. The minimal successful radius is recorded as the distance to the decision boundary for that sample, and $S(f)$ is reported as the average over the dataset.

**Normalized Co-Stability.** The empirical co-stability $S^*(g)$ is computed via the multi-class margin

$$g_j(x) - \max_{i \neq j} g_i(x), \qquad j = \arg\max_i g_i(x),$$

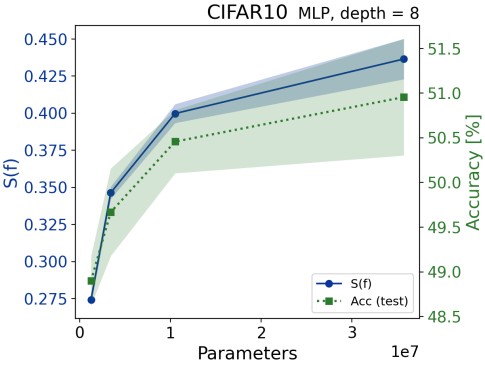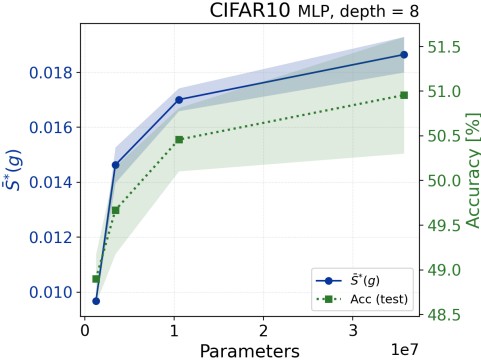

Figure 1: 8-layer MLPs on CIFAR-10. Class stability $S(f)$ (left) and normalized co-stability $S^*(g)/L(g)$ (right) as a function of model size. The configuration with $w = 128$ is excluded since it failed to attain 99% training accuracy after 400 epochs.

averaged over the dataset; see Appendix F for details about the multi-class setting. We estimate the Lipschitz constant $L(g)$ using the efficient ECLIPSE method (Xu & Sivaranjani, 2024), and report the normalized ratio $S^*(g)/L(g)$ as a function of model size.

**Results.** Figure 1 illustrates the behaviour for MLPs on CIFAR-10. Both stability measures increase with width and follow the same qualitative trend as test accuracy. In contrast, standard weight norms (or their inverses) exhibit a different width-scaling and do not track test accuracy comparably. The observed saturation of (normalized co-)stability aligns with theoretical intuition: the Bayes classifier admits a finite (normalized co-)stability level, and pushing beyond this level necessarily reduces accuracy – an instance of the robustness/accuracy trade-off extensively discussed in the literature (Zhang et al., 2019; Tsipras et al., 2019; Béthune et al., 2022). Accordingly, we expect stability to plateau once models approach the Bayes decision boundary. For CIFAR-10, test accuracy in our experiments remains around 50%, which is far below the Bayes optimal. Nevertheless, the same reasoning applies relative to the best classifier achievable within the restricted MLP hypothesis class considered here.

CNN experiments (Figure 2) display the same qualitative scaling of class stability with model size.

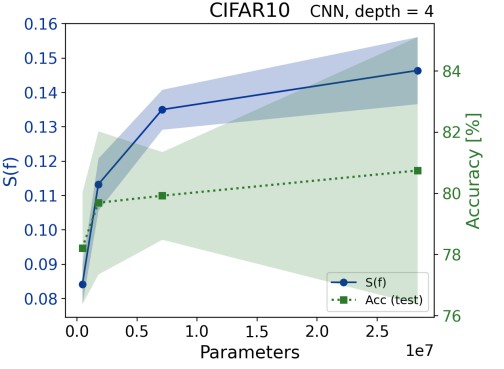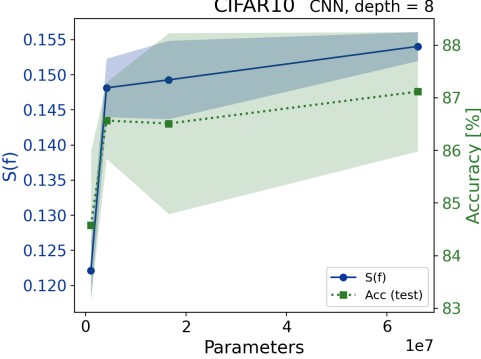

Figure 2: Class stability $S(f)$ for 4- and 8-layer CNNs trained on CIFAR-10.

The Heaviside MLP experiments (Figure 3) show that the observed stability scaling persists even for discontinuous score functions. This suggests that the Lipschitz assumption employed to extend Theorem 4 to infinite function classes is primarily technical rather than intrinsic to the stability–model size relationship.

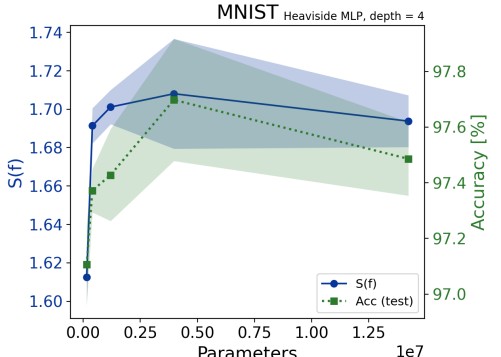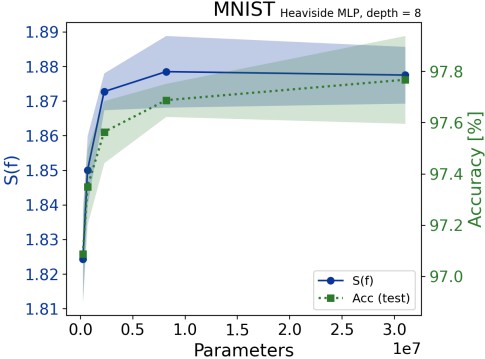

Figure 3: Class stability $S(f)$ for 4-layer and 8-layer Heaviside-activation MLPs trained on MNIST.

# 7 DISCUSSION AND FUTURE WORK

Our results identify class stability and its codomain analogue, normalized co-stability, as principled quantities linking overparameterization, generalization, and robustness for discontinuous classifiers. While we provide geometric laws of robustness for finite and infinite hypothesis classes, and our experiments support their validity, several directions remain open.

**Empirical directions.** Computing class stability $S(f)$ and Lipschitz constants $L(g)$ of neural networks is NP-hard (Katz et al., 2017; Weng et al., 2018; Scaman & Virmaux, 2019), limiting the direct use of (normalized co-)stability in training. However, practical relaxations exist: normalized co-stability underlies *Lipschitz margin training* (Tsuzuku et al., 2018), while input-space stability is related to adversarial training (Madry et al., 2018; Goodfellow et al., 2015). Biasing optimization explicitly toward (co-)stable solutions is therefore a promising empirical direction.

Another direction is to investigate how isoperimetric structure and related concentration phenomena manifest in real datasets and how they affect classifier stability. To isolate the role of the isoperimetric scale in a controlled setting, we consider a toy experiment with synthetic Gaussian data. Increasing the variance, and thus reducing concentration, leads to a deterioration of the stability–width scaling (Figure 4, Appendix A). The question of whether real data exhibit comparable geometric structure connects naturally to the manifold hypothesis: normalized Riemannian volume measures on positively curved manifolds satisfy isoperimetric inequalities. This raises the question of whether robustness laws fail empirically when the effective dimension of the data manifold is small.

**Theoretical directions.** Our framework also motivates the exploration of alternative geometric complexity measures. Do quantities such as sharpness of the loss landscape obey robustness laws analogous to those for (normalized co-)stability? Another question concerns sufficiency: we establish that overparameterization is necessary for generalization, but is it also sufficient under suitable optimization? Bombari et al. (2023) prove sufficiency for Lipschitz regression in the NTK regime but show that it fails for a random features model. Extending such results to discontinuous classifiers may reveal qualitative differences.

Finally, the role of implicit bias remains unclear. Does gradient descent or SGD exhibit a bias toward classifiers with higher (normalized co-)stability, as suggested by analogous results on region counts (Li et al., 2025)? Establishing such a bias would explain why stable solutions emerge in practice.

Overall, our findings suggest that stability-based laws capture a core structural constraint of modern overparameterized learning. Developing efficient estimators, stronger empirical validation, and deeper theoretical connections (e.g., with sharpness and optimization bias) are promising next steps toward a unified understanding of generalization and robustness.

ACKNOWLEDGMENTS

Jonas von Berg, Massimiliano Datres and Gitta Kutyniok acknowledge support by the project "Next Generation AI Computing (gAIn)," funded by the Bavarian Ministry of Science and the Arts and the Saxon Ministry for Science, Culture, and Tourism as well as by the Hightech Agenda Bavaria.

Jonas von Berg and Gitta Kutyniok are also grateful for partial support from the Konrad Zuse School of Excellence in Reliable AI (DAAD).

Additionally, Jonas von Berg, Gitta Kutyniok, Adalbert Fono, Massimiliano Datres and Sohir Maskey acknowledge support by the Munich Center for Machine Learning (MCML).

Gitta Kutyniok furthermore acknowledges support by the German Research Foundation under Grants DFG-SPP-2298, KU 1446/31-1 and KU 1446/32-1, and by the Bavarian Ministry for Digital Affairs.

ETHICS STATEMENT

This work focuses on the theoretical analysis of generalization in machine learning and does not involve experiments on human subjects, sensitive personal data, or applications with direct societal risks. The datasets referenced are publicly available, and no private or restricted data was used. Potential ethical concerns related to misuse are minimal, as the contributions are mainly theoretical and methodological.

ACKNOWLEDGMENT OF LLM USE

We explicitly acknowledge that large language models (LLMs) were used solely for polishing code, improving sentence clarity, and refining grammar. They were not used for generating research ideas, proofs, or results.

REPRODUCIBILITY STATEMENT

We have taken multiple steps to ensure reproducibility of our results. All theoretical claims are accompanied by rigorous proofs, presented in detail in the appendix. Assumptions underlying the theorems are explicitly stated, and definitions are given in full to allow independent verification.

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

# A THE NEED FOR ISOPERIMETRY

## A.1 THEORETICAL JUSTIFICATION

Concentration inequalities are essential tools in high-dimensional probability theory, providing bounds on the tail behavior of random variables. Next, we outline the key strategy from Bubeck & Sellke (Bubeck & Sellke, 2021) for proving the law of robustness for regression, highlighting the importance of an additional assumption on the measure $\mu$. The authors employ the Lipschitz constant of a function as a measure of robustness, where a small Lipschitz constant (i.e., $\approx 1$) of the realization indicates a robust model. The basic idea is to leverage the Lipschitz continuity of functions $f : \mathcal{X} \to \mathbb{R}$ in conjunction with isoperimetric inequalities to bound the probability

$$\mathbb{P}(\exists f \in \mathcal{F} : \hat{R}_\ell(f) \approx 0 \ \wedge \ L(f) \leq L_*) < \delta. \tag{9}$$

That is, we aim to bound the probability of observing a model that is both robust (i.e., has a small Lipschitz constant $L(f)$) and fits the data well (i.e., $\hat{R}(f) \approx 0$, meaning it nearly interpolates). By contraposition, this implies that with probability at least $1 - \delta$, the following holds for all $f \in \mathcal{F}$:

$$\hat{R}_\ell(f) \approx 0 \implies L(f) > L_*(p, n, d). \tag{10}$$

Here, $L_*(p, n, d)$ is an algebraic function of the number of parameters $p \approx \log |\mathcal{F}|$ (see the paragraph below Theorem 4 for details), the number of training samples $n$, and the input dimension $d$. It satisfies $L_*(p, n, d) \gg 1$ in the non-overparameterized regime $p \approx n$, thereby implying non-robust behavior.

A key ingredient in Bubeck & Sellke (2021) for proving (a variant of) Equation 9 is the isoperimetry assumption on the measure $\mu$. Isoperimetry, originating in geometry, provides an upper bound on a set's volume in terms of its boundary's surface area. In high dimensions, the principle of isoperimetry induces a concentration of measure, where the measure of the $\varepsilon$-neighborhood $A_\varepsilon$ of any set $A$ with $\mu(A) > 0$ has measure $\mu(A_\varepsilon) \to 1$, and the complementary measure decays in the order of $\exp(-d\varepsilon^2)$. This is equivalent to the sub-Gaussian behavior of every bounded Lipschitz-continuous function as stated in Definition 3, yielding a concentration property for $|f(x) - \mathbb{E}(f)|$ that depends on the Lipschitz constant $L(f)$.

The induced concentration property allows us to bound the probability in Equation 9, leveraging the intuition that a smaller Lipschitz constant limits the function's capacity to align with random labels. However, it is important to note that Equation 10 provides information about robustness within $\mathcal{F}$ only if

$$\mathbb{P}(\nexists f \in \mathcal{F} : \hat{R}_\ell(f) \approx 0) \leq 1 - \delta \iff \mathbb{P}(\exists f \in \mathcal{F} : \hat{R}_\ell(f) \approx 0) \geq \delta.$$

Otherwise, the implication becomes vacuous, as almost no function in $\mathcal{F}$ generalizes well, i.e., achieves near-zero empirical risk, to begin with. Without imposing any assumptions on $\mu$, Hoeffding's inequality already suffices to derive a Lipschitz-independent bound for any function $f : \mathcal{X} \to [-1, 1]$:

$$\mathbb{P}(|f(x) - \mathbb{E}(f)| \geq t) \leq 2 \exp\left(-\frac{t^2}{2}\right) \quad \forall t > 0. \tag{11}$$

Thus, to ensure that the probability in Equation 9 remains below $\delta$ while simultaneously allowing for $\mathbb{P}(\exists f \in \mathcal{F} : \hat{R}_\ell(f) \approx 0) > \delta$, any concentration inequality relying on the Lipschitz constant must exhibit a sufficiently fast decay (in comparison with Equation 11) in the regime $L(f) \gtrsim 1$. This is necessary to yield a non-vacuous bound in Equation 10, which allows to assess robustness by the increase of the minimal Lipschitz constant $L_*$ even for $L_* > 1$.

For instance, McDiarmid's inequality applied to Lipschitz functions yields a tail bound of the order $\exp(-\frac{2t^2}{\text{diam}(\mathcal{X})^2 L(f)^2})$, which is insufficient as it decays faster than the Hoeffding bound only for $L(f) < 2/\text{diam}(\mathcal{X})$, i.e., at least $\text{diam}(\mathcal{X}) < 2$ is required to include the (relevant) range $L(f) > 1$ of Lipschitz constants. This indicates that a certain restriction of the admissible measures is indeed necessary to obtain non-vacuous statements, i.e., they can not be derived in full generality.

Notably, the $c$-isoperimetry condition in Equation 3 leads to a faster decay than the Hoeffding bound in Equation 11 when $L(f) < \sqrt{dc^{-1}}$, making it effective for functions with moderate Lipschitz constants in high-dimensional settings. Our goal is to generalize this strategy to handle discontinuous functions, addressing the inherent challenges of classification tasks.

## A.2 EXPERIMENTS ON ISOPERIMETRY

To empirically investigate the role of the isoperimetric assumption in the relationship between accuracy and stability, we conduct experiments on Gaussian data with increasing variance. We train 4-layer MLPs of varying width $w \in \{128, 256, 512, 1024, 2048\}$ on a linear classification task. To ensure comparable training conditions across widths, we first determine the number of epochs required for the smallest model to reach $99\%$ training accuracy and then train all wider models for the same number of epochs. Under this training schedule, all models achieve $99\%$ training accuracy. In this setting, the isoperimetric constant scales with the product of the variance and the ambient dimension $d = 784$, as follows from standard Gaussian concentration inequalities.

The first two plots in Figure 4 correspond to the regime in which the isoperimetric scale $c \sim \sigma^2 d$ remains $\mathcal{O}(1)$. In this setting, stability increases with width and subsequently plateaus, mirroring the behaviour observed on MNIST and CIFAR-10. For larger values of $c$, this plateau structure disappears: stability becomes non-monotonic and decreases for sufficiently large widths.

These results indicate that the beneficial effect of overparameterization on stability is regime-dependent and can be disrupted when the underlying isoperimetric scale becomes large. However, confirming this behavior would require experiments across a broader range of network widths and systematic evaluation on more complex datasets.

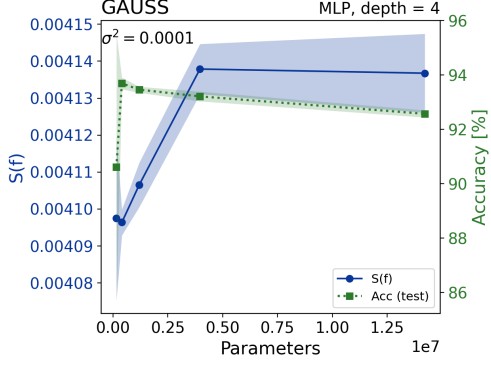
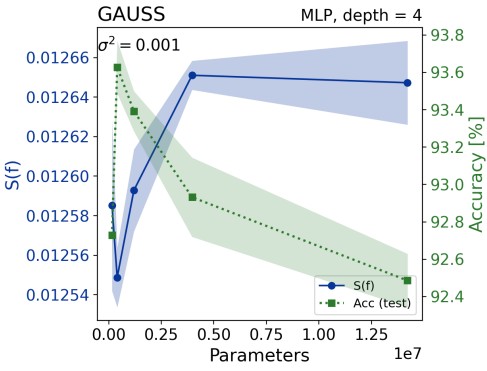

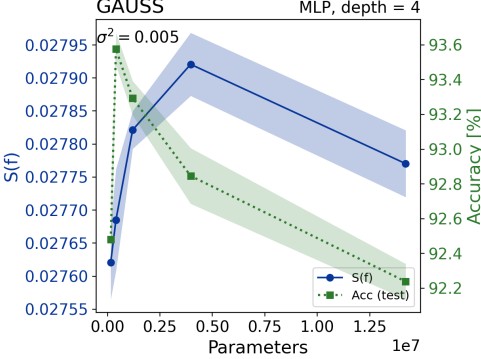
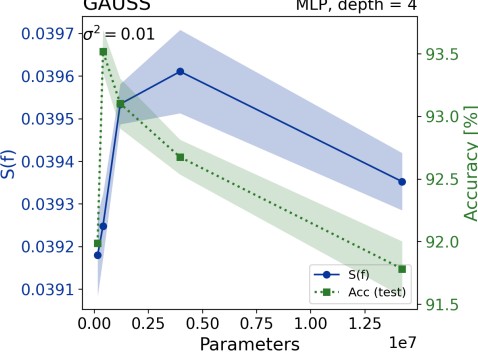

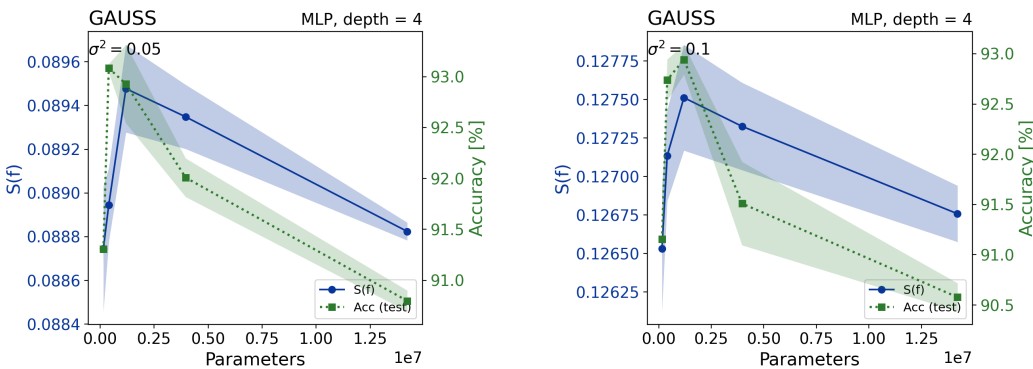

Figure 4: Class stability for 4-layer MLPs trained on Gaussian toy-data with different variances.

## B   THE SIGNED DISTANCE FUNCTION (REMARK 2)

We collect the main properties of the signed distance function

$$
d_f(x) := \begin{cases} d(x, f^{-1}(\{-1\})), & \text{if } f(x) = 1, \\ -d(x, f^{-1}(\{1\})), & \text{if } f(x) = -1, \end{cases}
$$

where $d(x, A) := \inf_{y \in A} \|x - y\|_2$.

**Lemma 17.** *Let $\mathcal{X} \subset \mathbb{R}^d$ be bounded and path-connected, and let $f : \mathcal{X} \to \{-1, 1\}$. Then the signed distance function $d_f$ is 1-Lipschitz.*

This is a classical fact, a special case of the Eikonal equation. For completeness, we include a direct proof inspired by Liu & Hansen (2024, Prop. 7.5).

*Proof.* **Case 1:** $f(x) = f(y)$. Assume w.l.o.g. $f(x) = f(y) = 1$. Let $(z_n)_n$ be a sequence in $f^{-1}(\{-1\})$ with $|d(y, z_n) - d_f(y)| \le \frac{1}{n}$. Then

$$
\begin{aligned}
d_f(x) &= d(x, f^{-1}(\{-1\})) \\
&\le d(x, z_n) \\
&\le \|x - y\|_2 + d(y, z_n) \\
&\le \|x - y\|_2 + d_f(y) + \tfrac{1}{n}.
\end{aligned}
$$

Letting $n \to \infty$ and exploiting symmetry yields $|d_f(x) - d_f(y)| \le \|x - y\|_2$.

**Case 2:** $f(x) \neq f(y)$. Assume w.l.o.g. $f(x) = 1$, $f(y) = -1$. Consider the line segment $L = \{(1 - t)x + ty : t \in [0, 1]\} \subset \mathcal{X}$ and define

$$
\begin{aligned}
w_1 &= (1 - t_1)x + t_1 y, & t_1 &:= \inf\{t : f((1 - t)x + ty) = -1\}, \\
w_2 &= (1 - t_2)x + t_2 y, & t_2 &:= \sup\{t : f((1 - t)x + ty) = 1\}.
\end{aligned}
$$

Path-connectedness ensures $t_1 \le t_2$, otherwise the midpoint between $w_1$ and $w_2$ would be labeled both 1 and $-1$, a contradiction.

Thus,

$$
\begin{aligned}
|d_f(x) - d_f(y)| &= d(x, f^{-1}(\{-1\})) + d(y, f^{-1}(\{1\})) \\
&\le \|x - w_1\|_2 + \|y - w_2\|_2 \\
&\le \|x - y\|_2.
\end{aligned}
$$

$\square$

**Lemma 18.** *Let $\mathcal{X} \subset \mathbb{R}^d$ and $f : \mathcal{X} \to \{-1, 1\}$ with $f^{-1}(\{1\})$ closed. Then $f$ can be represented as*

$$
f(x) = \operatorname{sgn}(d_f(x)),
$$

*where we adopt the convention $\operatorname{sgn}(0) = 1$.*

*Proof.* If $d_f(x) \neq 0$, the claim follows directly from the definition of $d_f$. If $d_f(x) = 0$, then $x \in f^{-1}(\{1\})$ by closedness, so $f(x) = 1 = \text{sgn}(0)$. $\qquad \square$

**Remark 19.** *Lemma 18 justifies the representation $f = \text{sgn} \circ d_f$ used in the proof of Theorem 4. This link between classifiers and their signed distance functions is what allows stability arguments to be combined with smoothness-based tools.*

## C   PROOF OF THE RADEMACHER BOUND (THEOREM 4)

In the regression setting, it is natural to restrict to Lipschitz-continuous regressors, as many commonly used model classes (e.g., neural networks with bounded weights and Lipschitz activations) satisfy this property. This assumption enables quantitative statements about expected and achievable robustness. In contrast, this approach is not meaningful anymore in the classification setting as the considered classifiers are (except for trivial cases) discontinuous by design, i.e., they can not be captured by a finite Lipschitz constant. Thus, statements about the robustness of classification models can not be derived via Lipschitz constants. This motivates the use of class stability as a replacement measure in the classification setting, which, however, is (inversely) related to Lipschitzness as highlighted and exploited in the subsequent proof of Theorem 4. For convenience, we repeat the statement with the corresponding assumptions.

(H1)  $(\mathcal{X}, \mu)$ is a probability space with bounded sample space $\mathcal{X}$ and c-isoperimetric measure $\mu$;

(H2)  the considered hypothesis class $\mathcal{F}$ of classifiers $f : \mathcal{X} \to \{-1, 1\}$ is finite, that is $|\mathcal{F}| < \infty$.

**Theorem** (Rademacher Bound). *Suppose Assumptions (H1) and (H2) hold, and that $\min_{f \in \mathcal{F}} S(f) > S > 0$ with $\log |\mathcal{F}| \geq n$.*

1. *The empirical Rademacher complexity satisfies*

$$\mathcal{R}_{n,\mu}(\mathcal{F}) \leq K_1 \max \left\{ \frac{1}{\sqrt{n}}, \ \frac{\sqrt{c}}{S} \cdot \frac{\log |\mathcal{F}|}{n\sqrt{d}} \right\}, \tag{12}$$

   *for an absolute constant $K_1 > 0$.*

2. *If, in addition, $f^{-1}(\{1\})$ is closed and $\mathcal{X}$ path connected, the bound sharpens to*

$$\mathcal{R}_{n,\mu}(\mathcal{F}) \leq K_2 \max \left\{ \frac{1}{\sqrt{n}}, \ \frac{\sqrt{c}}{S} \sqrt{\frac{\log |\mathcal{F}|}{nd}}, \ 2 \exp\left( -\frac{dS^2}{8c} \right) \right\}, \tag{13}$$

   *for another absolute constant $K_2 > 0$.*

*Proof.*  : **1.**  To begin, we explore the relationship between two measures of robustness: the Lipschitz constant $L(f)$ and the class stability $S(f)$ of a $f \in \mathcal{F}$ on the set

$$A_t(f) := \{x \in \mathcal{X} : h_f(x) > S(f) - t\} \quad \text{for } 0 \leq t \leq S(f).$$

Observe that for $x_1 \in A_t(f)$ and $x_2 \in \mathcal{X}$

$$|f(x_1) - f(x_2)| \leq \begin{cases} 0, & \text{if } f(x_1) = f(x_2) \\ 2 \cdot \overbrace{\frac{\|x_1 - x_2\|}{S(f) - t}}^{\geq 1}, & \text{if } f(x_1) \neq f(x_2) \end{cases} \leq \frac{2}{S(f) - t} \|x_1 - x_2\|,$$

i.e., $f$ is $\frac{2}{S(f)-t}$-Lipschitz on $A_t(f)$ and, therefore, according to the assumption $S(f) > S$, any $f \in \mathcal{F}$ is at most $\frac{2}{S-t}$-Lipschitz on $A_t(f)$. Our strategy now is to apply the Rademacher bound based on Lipschitz functions of Bubeck & Sellke in Bubeck & Sellke (2021) to the restriction $f_{|A_t(f)}$, and additionally exploit isoperimetry to control the measure of the complement $A_t(f)^c$. We rely on two key facts:

- **Fact 1**: Every Lipschitz continuous function $g : A \rightarrow \mathbb{R}$, defined on a subset $A \subset \mathcal{X}$ of a metric space, can be extended to a function $G_g : \mathcal{X} \rightarrow \mathbb{R}$, preserving the same Lipschitz constant (McShane (1934), Kirszbraun (1934)). $\implies$ *This allows us to apply isoperimetry and thereby the result in (Bubeck & Sellke, 2021, Lemma 4.1) to the $\frac{2}{S-t}$-Lipschitz extension $F_f$ of $f_{|A_t(f)}$ (by w.l.o.g. restricting its codomain to $[-1,1]$) to obtain*

$$\frac{1}{n}\mathbb{E}^{\sigma_i, x_i}\left[\sup_{f \in \mathcal{F}}\left|\sum_{i=1}^{n}\sigma_i F_f(x_i)\right|\right] \leq C_1 \frac{1}{\sqrt{n}} + C_2 \frac{1}{S-t}\sqrt{\frac{c \log |\mathcal{F}|}{nd}}$$

for some absolute constants $C_1, C_2 > 0$.

- **Fact 2**: The margin $h_f(x) : \mathcal{X} \rightarrow \mathbb{R}$, is 1-Lipschitz continuous with respect to the $\ell_2$-norm ((Liu & Hansen, 2024, Prop. 7.5). $\implies$ *This allows us to control $\mathbb{P}(A_t(f)^c)$ via isoperimetry:*

$$\mathbb{P}(A_t(f)^c) = \mathbb{P}(\overbrace{S(f)}^{=\mathbb{E}[h_f]} - h_f(x) \geq t) \leq \exp\left(-\frac{dt^2}{2cL(h_f)^2}\right) = \exp\left(-\frac{dt^2}{2c}\right). \quad (14)$$

Via Fact 1, we can bound the Rademacher complexity by

$$\mathcal{R}_{n,\mu}(\mathcal{F}) = \frac{1}{n}\mathbb{E}^{\sigma_i, x_i}\left[\sup_{f \in \mathcal{F}}\left|\sum_{i=1}^{n}\sigma_i f(x_i)\right|\right]$$

$$\leq \frac{1}{n}\mathbb{E}^{\sigma_i, x_i}\left[\sup_{f \in \mathcal{F}}\left|\sum_{i=1}^{n}\sigma_i F_f(x_i)\right|\right] + \frac{1}{n}\mathbb{E}^{\sigma_i, x_i}\left[\sup_{f \in \mathcal{F}}\left|\sum_{i=1}^{n}\sigma_i (f - F_f)(x_i)\right|\right]$$

$$\leq C_1 \frac{1}{\sqrt{n}} + C_2 \frac{1}{S-t}\sqrt{\frac{c \log |\mathcal{F}|}{nd}} + \frac{1}{n}\mathbb{E}^{\sigma_i, x_i}\left[\sup_{f \in \mathcal{F}}\left|\sum_{i=1}^{n}\sigma_i (f - F_f)(x_i)\right|\right]. \quad (15)$$

To control the last term, we subdivide $\mathcal{X}^n$ into subsets on which specific samples achieve a minimum margin. To that end, we fix $t = \frac{S}{2}$ (the exact value is not crucial since it will be subsumed into the absolute constants) and define, for $I \subset [n]$,

$$A^I(f) = A^I_{\frac{S}{2}}(f) := \left\{x \in \mathcal{X}^n : i \in I \iff h_f(x_i) \geq \frac{S}{2}\right\}.$$

Note, that $A^{[n]}(f) = A_{\frac{S}{2}}(f)^n$ and $\cup_{I \in \mathcal{P}([n])} A^I(f)$ is a disjoint partition of $\mathcal{X}^n$. Thus, applying a union bound yields for $r > 0$

$$\mathbb{P}\left(\sup_{f \in \mathcal{F}}\left|\sum_{i=1}^{n}\sigma_i (f - F_f)(x_i)\right| > r\right) \leq \sum_{f \in \mathcal{F}}\sum_{I \in \mathcal{P}([n])}\mathbb{P}\left(\left|\sum_{i=1}^{n}\sigma_i (f - F_f)(x_i)\right| > r \wedge x \in A^I(f)\right)$$

$$= \sum_{f \in \mathcal{F}}\sum_{I \in \mathcal{P}([n])}\mathbb{P}\left(\left|\sum_{i=1}^{n}\sigma_i (f - F_f)(x_i)\right| > r \,\middle|\, x \in A^I(f)\right)\mathbb{P}(A^I(f)). \quad (16)$$

We make the following observations:

- By construction $F_f = f$ holds on $A^I(f)$ for all $f \in \mathcal{F}$.

- As a mean-zero and bounded random variable with range $[-2, 2]$, $\sigma_i(F_f - f)(x_i)$ is (via Hoeffding's inequality) subgaussian with variance proxy $\frac{(2-(-2))^2}{4} = 4$ for every $i \in [n], f \in \mathcal{F}$.

Using the fact that the sum of $k$ independent subgaussian random variables with variance proxy $\sigma^2$ is itself subgaussian with variance proxy $k\sigma^2$ (Rigollet & Hütter, 2023), we obtain for every $I \subsetneq [n]$ (the case $I = [n]$ being trivial) that

$$\mathbb{P}\left(\left|\sum_{i=1}^{n}\sigma_i(f-F_f)(x_i)\right| > r \,\Big|\, x \in A^I(f)\right) \leq \mathbb{P}\left(\left|\sum_{i\in I^c}\sigma_i(f-F_f)(x_i)\right| > r \,\Big|\, x \in A^I(f)\right)$$

$$\leq 2\exp\left(-\frac{r^2}{2\cdot 4(n-|I|)}\right).$$

On the other hand, we get for $I \subset [n]$ via Equation 14 that

$$\mathbb{P}\left(A^I(f)\right) \leq \mathbb{P}\left(\forall j \in I^c:\ x_j \in A_{\frac{S}{2}}(f)^c\right) = \mathbb{P}\left(x \in A_{\frac{S}{2}}(f)^c\right)^{n-|I|} \leq \exp\left(-\frac{dS^2}{2^3 c}\right)^{n-|I|}.$$

Inserting in Equation 16 and replacing the constants independent of the parameters of interest $(n, |\mathcal{F}|, d, r, S, \text{ and } |I|)$ by $c_1, c_2 > 0$ then gives

$$\mathbb{P}\left(\sup_{f\in\mathcal{F}}\left|\sum_{i=1}^{n}\sigma_i(f-F_f)(x_i)\right| > r\right) \leq \sum_{f\in\mathcal{F}}\sum_{I\in\mathcal{P}([n])\setminus[n]} 2\exp\left(-\frac{r^2 c_1}{n-|I|}\right)\exp\left(-\frac{(n-|I|)dS^2 c_2}{c}\right).$$

To simplify the above expression, we want to find the maximal term in the sum and use this worst case as an upper bound over all terms in the sum. To that end, we introduce $g : [0, n) \to \mathbb{R}_+$ by

$$g(x) = \frac{r^2 c_1}{n-x} + \frac{1}{c}(n-x)S^2 dc_2,$$

aiming to find its minima, which correspond to an upper bound on the sought worst-case term. Differentiating $g$ yields the extrema

$$g'(x) = \frac{r^2 c_1}{(n-x)^2} - \frac{1}{c}S^2 dc_2 \overset{!}{=} 0$$

$$\implies x_{+/-} = n \pm \frac{r}{S}\sqrt{\frac{c_1 c}{c_2 d}} =: n \pm \alpha(r) \tag{17}$$

We calculate the second derivatives to be $g''(x_-) > 0$ and $g''(x_+) < 0$, thus only $x_-$ is a minimum. Now, there are two cases associated with the location of $x_-$ (taking into account that $\alpha(r) > 0$ for every $r > 0$).

- **Case I**: $\alpha(r) \leq n$.
  Then, $x_-$ is a valid minimum in the considered range and therefore

  $$\mathbb{P}\left(\sup_{f\in\mathcal{F}}\left|\sum_{i=1}^{n}\sigma_i(f-F_f)(x_i)\right| > r\right)$$

  $$\leq \sum_{f\in\mathcal{F}}\sum_{I\in\mathcal{P}([n])\setminus[n]} 2\exp\left(-\frac{r^2 c_1}{\alpha(r)}\right)\exp\left(-\frac{\alpha(r)dS^2 c_2}{c}\right)$$

  $$\leq 2|\mathcal{F}|2^n \exp\left(-2rS\sqrt{\frac{dc_2 c_1}{c}}\right) := \mathbb{P}_{(I)}(r).$$

- **Case II**: $\alpha(r) > n$.
  Then, $x_- < 0$ is outside of the domain of $g$. However, the derivative satisfies $g'(x) > 0$ for any $0 \leq x < n$ since $x_+ > n$. Therefore, $g$ necessarily takes its minimal value at $x = 0$ so that

  $$\mathbb{P}\left(\sup_{f\in\mathcal{F}}\left|\sum_{i=1}^{n}\sigma_i(f-F_f)(x_i)\right| > r\right)$$

  $$\leq \sum_{f\in\mathcal{F}}\sum_{I\in\mathcal{P}([n])\setminus[n]} 2\exp\left(-\frac{r^2 c_1}{n}\right)\exp\left(-\frac{ndS^2 c_2}{c}\right)$$

  $$\leq 2|\mathcal{F}|2^n \exp\left(-\frac{r^2 c_1}{n}\right)\exp\left(-\frac{ndS^2 c_2}{c}\right) =: \mathbb{P}_{(II)}(r).$$

Using Equation 17, the condition $\alpha(r) > n$ is equivalent to $r > nS\sqrt{\frac{c_2 d}{c_1 c}}$. In this range, we have $\mathbb{P}_{(II)}(r) \leq \mathbb{P}_{(I)}(r)$ since

$$\mathbb{P}_{(II)}\left(nS\sqrt{\frac{c_2 d}{c_1 c}}\right) = 2|\mathcal{F}|2^n \exp\left(-2nS^2 dc^{-1}c_2\right) = \mathbb{P}_{(I)}\left(nS\sqrt{\frac{c_2 d}{c_1 c}}\right)$$

and one verifies that $\mathbb{P}_{(II)}(r)$ decays faster than $\mathbb{P}_{(I)}(r)$ when further increasing $r$. Therefore, we conclude that for all $r > 0$

$$\mathbb{P}\left(\sup_{f \in \mathcal{F}}\left|\sum_{i=1}^{n} \sigma_i (f - F_f)(x_i)\right| > r\right) \leq \mathbb{P}_{(I)}(r) = 2|\mathcal{F}|2^n \exp\left(-2rS\sqrt{\frac{dc_2 c_1}{c}}\right). \quad (18)$$

Further rewriting the expression, distinguishing between two cases with respect to the magnitude of $|\mathcal{F}|2^n$ yields the upper bounds:

- **Case 1**: $|\mathcal{F}|2^n \leq \exp\left(rS\sqrt{\frac{dc_2 c_1}{c}}\right)$.

  We immediately obtain via Equation 18 that

  $$\mathbb{P}\left(\sup_{f \in \mathcal{F}}\left|\sum_{i=1}^{n} \sigma_i (f - F_f)(x_i)\right| > r\right) \leq 2|\mathcal{F}|2^n \exp\left(-2rS\sqrt{\frac{dc_2 c_1}{c}}\right)$$

  $$\leq 2\exp\left(-rS\sqrt{\frac{dc_2 c_1}{c}}\right)$$

  $$\leq 2\exp\left(-\underbrace{\frac{2}{3\log(|\mathcal{F}|2^n)}}_{<1} rS\sqrt{\frac{dc_2 c_1}{c}}\right).$$

- **Case 2**: $|\mathcal{F}|2^n > \exp\left(rS\sqrt{\frac{dc_2 c_1}{c}}\right)$.

  In this case, the probability is trivially bounded by

  $$\mathbb{P}\left(\sup_{f \in \mathcal{F}}\left|\sum_{i=1}^{n} \sigma_i (f - F_f)(x_i)\right| > r\right) \leq 1 < 2\exp\left(-\frac{2}{3}\right) < 2\exp\left(-\frac{2}{3}\frac{rS\sqrt{\frac{dc_2 c_1}{c}}}{\underbrace{\log(|\mathcal{F}|2^n)}_{<1}}\right)$$

Putting both cases together, we proved that for all $r > 0$

$$\mathbb{P}\left(\sup_{f \in \mathcal{F}}\left|\sum_{i=1}^{n} \sigma_i (f - F_f)(x_i)\right| > r\right) \leq 2\exp\left(-\frac{2S\sqrt{\frac{dc_2 c_1}{c}}}{3\log(|\mathcal{F}|2^n)}r\right).$$

This tail bound shows that $\sup_{f \in \mathcal{F}}|\sum_{i=1}^{n} \sigma_i (f - F_f)(x_i)|$ is sub-exponential. Since the expected value of any sub-exponential random variable is up to an absolute constant given by its sub-exponential norm, which corresponds (up to a constant) to the parameter $\frac{3\log(|\mathcal{F}|2^n)}{2S\sqrt{\frac{dc_2 c_1}{c}}}$ in the tail bound Vershynin (2018), we obtain for a constant $C_3 > 0$ that

$$\frac{1}{n}\mathbb{E}^{\sigma_i, x_i}\left[\sup_{f \in \mathcal{F}}\left|\sum_{i=1}^{n} \sigma_i (f - F_f)(x_i)\right|\right] \leq C_3 \frac{1}{S}\left(\frac{\log|\mathcal{F}| + n\log 2}{n\sqrt{\frac{d}{c}}}\right)$$

Finally, the desired bound on the Rademacher complexity follows via Equation 15:

$$\mathcal{R}_{n,\mu}(\mathcal{F}) = \frac{1}{n}\mathbb{E}^{\sigma_i, x_i}\left[\sup_{f \in \mathcal{F}}\left|\sum_{i=1}^{n} \sigma_i f(x_i)\right|\right]$$

$$\leq C_1 \frac{1}{\sqrt{n}} + C_2 \frac{1}{S}\sqrt{\frac{c\,\log|\mathcal{F}|}{nd}} + C_3 \frac{1}{S}\frac{\sqrt{c}\log|\mathcal{F}|}{n\sqrt{d}} + C_3 \frac{1}{S}\sqrt{\frac{c}{d}},$$

which, with the additional assumption $\log|\mathcal{F}| \geq n$, gives the result in 1.

**2.** By Lemma 18, every $f$ admits the representation $f = \mathrm{sgn} \circ d_f$. This lets us follow the infinite-class analysis (presented in detail in the proof of Theorem 13), without the $\varepsilon$-net step in Equation 22. From Lemma 17, $d_f$ is 1-Lipschitz, i.e., $L(d_f) = 1$ under the given conditions. Furthermore, recalling the co-stability definition we get

$$S^*(d_f) = \mathbb{E}[|d_f|] = \mathbb{E}[h_f] = S(f).$$

Plugging this into the general bound in Equation 8 gives the result. $\square$

## C.1 COMPARISON TO STANDARD BOUND WITHOUT ACCOUNTING FOR STABILITY

Note that the crucial expectation in the derivation, i.e., the last term in Equation 15, can be treated without linking it to the minimum class stability. Indeed, the expectation of the maximum of $N$ subgaussians $X_1, \ldots, X_N$ with variance proxy $\sigma^2$ scales as

$$\mathbb{E}\left[\max_{1 \leq i \leq N} |X_i|\right] \leq \sigma\sqrt{2\log(2N)}, \tag{19}$$

see for instance Rigollet & Hütter (2023). Hence, in our case, as $\sigma_i(f - F_f)(x_i)$ is subgaussian with variance proxy 4 and therefore $\sum_{i=1}^{n} \sigma_i(f - F_f)(x_i)$ is subgaussian with variance proxy $4n$, we obtain

$$\frac{1}{n}\mathbb{E}^{\sigma_i, x_i}\left[\sup_{f \in \mathcal{F}}\left|\sum_{i=1}^{n} \sigma_i(f - F_f)(x_i)\right|\right] \leq \frac{1}{n}2\sqrt{n}\sqrt{2\log(2|\mathcal{F}|)} \leq C_4\left(\sqrt{\frac{1}{n}} + \sqrt{\frac{\log|\mathcal{F}|}{n}}\right).$$

for some absolute constant $C_4 > 0$. Neglecting the constants, this leads to the following comparison to our bound in Equation 5:

$$\frac{\sqrt{c}}{S}\sqrt{\frac{p}{nd}} \leq \sqrt{\frac{\log|\mathcal{F}|}{n}} \quad \Longleftrightarrow \quad S \geq \sqrt{\frac{c}{d}}.$$

Thus, under the isoperimetry condition, our bound improves on the standard Rademacher complexity estimate whenever the class stability $S$ exceeds $\sqrt{c/d}$, a mild requirement in high-dimensional settings.

## D PROOF OF THE LAW OF ROBUSTNESS (COROLLARY 6)

Next, we provide the proof of Corollary 6, which we repeat for convenience.

**Theorem** (Law of Robustness for Discontinuous Functions). *Assume (H1), (H2), and the additional conditions in 2. of Theorem 4 hold. Let $p := \log|\mathcal{F}| \geq n$. Fix $\varepsilon, \delta \in (0, 1)$ and consider the 0-1 loss $\ell_{0\text{--}1}$. There exists an absolute constant $K > 0$ such that, if*

1. *the minimal risk $R^* := \min_{f \in \mathcal{F}} R_{0\text{--}1}(f)$ satisfies $R^* \geq \varepsilon$, and*

2. *the sample size $n$ is large enough to ensure (i) $\frac{K}{\sqrt{n}} < \frac{\varepsilon}{3}$ and (ii) $\sqrt{\frac{2\log(2/\delta)}{n}} < \frac{\varepsilon}{2}$,*

*then with probability at least $1 - \delta$ (over the sample), the following holds uniformly for all $f \in \mathcal{F}$:*

$$\hat{R}_{0\text{--}1}(f) \leq R^* - \varepsilon \quad \Longrightarrow \quad S(f) < \max\left\{\frac{3K}{\varepsilon}\sqrt{\frac{c\log|\mathcal{F}|}{nd}}, \sqrt{\frac{8c}{d}\log\left(\frac{6K}{\varepsilon}\right)}\right\}.$$

*Proof.* Let $K > 0$ be an absolute constant such that Equation 5 holds, and define the threshold stability

$$S_* = S_*(p, n, d, \varepsilon) := \max\left\{\frac{3K}{\varepsilon}\sqrt{\frac{c\log|\mathcal{F}|}{nd}}, \sqrt{\frac{8c}{d}\log\left(\frac{6K}{\varepsilon}\right)}\right\}.$$

Then, Theorem 4, together with condition $2(i)$, implies that

$$\mathcal{R}_{n,\mu}(\mathcal{F}_{S_*}) \leq K \max\left\{\frac{1}{\sqrt{n}}, \frac{\sqrt{c}}{S_*}\sqrt{\frac{\log|\mathcal{F}|}{nd}}, 2\exp\left(-\frac{dS_*^2}{8c}\right)\right\} \leq \varepsilon/3,$$

where $\mathcal{F}_{S_*} := \{f \in \mathcal{F} : S(f) \geq S_*\}$ is the subset of functions in $\mathcal{F}$ with stability at least $S_*$. Hence, applying the generalization inequality Equation 1, together with condition $2(ii)$, gives with probability $1 - \delta$:

$$\sup_{f \in \mathcal{F}_{S_*}} \left(R_{0\text{-}1}(f) - \hat{R}_{0\text{-}1}(f)\right) \leq 2\mathcal{R}_{n,\mu}(\ell_{0\text{-}1} \circ \mathcal{F}_{S_*}) + \sqrt{\frac{2\log(2/\delta)}{n}} \leq \mathcal{R}_{n,\mu}(\mathcal{F}_{S_*}) + \frac{\varepsilon}{2} < \varepsilon,$$

where we additionally used Equation 2 in the second step. In particular, we can bound the probability

$$\mathbb{P}(\forall f \in \mathcal{F}_{S_*} : \hat{R}_{0\text{-}1}(f) > R^* - \varepsilon) \geq \mathbb{P}(\forall f \in \mathcal{F}_{S_*} : R_{0\text{-}1}(f) - \hat{R}_{0\text{-}1}(f) < \varepsilon) \geq 1 - \delta,$$

where the first inequality follows from

$$R_{0\text{-}1}(f) - \hat{R}_{0\text{-}1}(f) < \varepsilon \overset{\text{condition 1.}}{\Longrightarrow} R^* - \hat{R}_{0\text{-}1}(f) < \varepsilon \implies \hat{R}_{0\text{-}1}(f) > R^* - \varepsilon.$$

Decomposing this probability into two disjoint events

$$1 - \delta \leq \mathbb{P}(\forall f \in \mathcal{F}_{S_*} : \hat{R}_{0\text{-}1}(f) > R^* - \varepsilon) = \mathbb{P}(\forall f \in \mathcal{F} : \hat{R}_{0\text{-}1}(f) > R^* - \varepsilon)$$
$$+ \mathbb{P}(\exists f \in \mathcal{F}_{S_*}^c : \hat{R}_{0\text{-}1}(f) \leq R^* - \varepsilon), \quad (20)$$

enables us to easily recognize that the expression exactly characterizes the probability that the following implication, and thereby the result, holds uniformly for all $f \in \mathcal{F}$:

$$\hat{R}_{0\text{-}1}(f) \leq R^* - \varepsilon \implies S(f) < S_*.$$

Indeed, the implication above holds if, for a given data sample $(x_i, y_i)_{i=1}^n$, either

- no function $f \in \mathcal{F}$ satisfies $\hat{R}_{0\text{-}1}(f) \leq R^* - \varepsilon$, or

- any such $f$ lies in $\mathcal{F}_{S_*}^c$, that is, $S(f) < S_*$,

which is the case with probability at least $1 - \delta$ due to Equation 20. $\square$

## E  PROOF OF RADEMACHER BOUND FOR INFINITE FUNCTION CLASSES (THEOREM 13)

Here we show how to extend the result for finite function classes to infinite function classes by a covering argument, where the Lipschitz continuity of the parameterization turns out to be crucial. Please find the exact statement about the Rademacher complexity of infinite function classes (of a certain form) below, after restating our new regularity hypothesis replacing (H2).

(H3)  The hypothesis class $\mathcal{F}$ is of the form $\mathcal{F} = \text{sgn} \circ \mathcal{G}$, where $\mathcal{G} = \{g_w : \mathcal{X} \to [-1, 1] : w \in \mathcal{W}\}$ is a parameterized class of Lipschitz continuous functions. The parameter space $\mathcal{W} \subset \mathbb{R}^p$ is bounded with $\text{diam}(\mathcal{W}) \leq W$, and the parameterization is Lipschitz continuous, i.e.,

$$\|g_{w_1} - g_{w_2}\|_\infty \leq J\|w_1 - w_2\|.$$

**Theorem.** *Under assumptions (H1) and (H3), suppose that $S^*(g) > S^* > 0$ and $L(g) \leq L$ for all $g \in \mathcal{G}$. Furthermore, assume that $p \geq n$. Then, for any covering precision $\tilde{\varepsilon} > 0$,*

$$\mathcal{R}_{n,\mu}(\mathcal{F}) \leq K \max\left\{\sqrt{\frac{1}{n}}, \frac{L}{S^*}\sqrt{\frac{p}{nd}}\sqrt{c\log(1 + 60WJ\tilde{\varepsilon}^{-1})}, 2\exp\left(-\frac{dS^{*2}}{8cL^2}\right), \frac{J}{S^*}\tilde{\varepsilon}\right\},$$

*where $K > 0$ is an absolute constant independent of $p, n, d, S^*, c, L, J, \tilde{\varepsilon}, W$.*

*Proof.* Given any discontinuous classifier $f_w = \text{sgn} \circ g_w$ for $g_w \in \mathcal{G}$, define its Lipschitz continuous approximation for $\gamma > 0$ as

$$F_{f_w} = \text{sgn}_\gamma \circ g_w,$$

where

$$\text{sgn}_\gamma(t) := \begin{cases} -1, & t \le -\gamma, \\ \frac{t}{\gamma}, & t \in [-\gamma, \gamma], \\ 1, & t \ge \gamma. \end{cases}$$

This approximation satisfies the useful property that both $F_{f_w}$ and the absolute difference $|f_w - F_{f_w}|$ are Lipschitz continuous in both the input space $\mathcal{X}$ and the weight space $\mathcal{W}$, with

$$L(|\text{sgn}_\gamma \circ g_w - \text{sgn} \circ g_w|) = L(\text{sgn}_\gamma \circ g_w) = \frac{L(g_w)}{\gamma}. \tag{21}$$

Following the same strategy as in the proof of Theorem 4 with Lipschitz continuous approximations introduced above (see Equation 15), coupled with a covering argument as in Bubeck & Sellke (2021), we obtain

$$\mathcal{R}_{n,\mu}(\mathcal{F}) = \frac{1}{n} \mathbb{E}^{\sigma_i, x_i} \left[ \sup_{f \in \mathcal{F}} \left| \sum_{i=1}^n \sigma_i f(x_i) \right| \right]$$

$$\le \frac{1}{n} \mathbb{E}^{\sigma_i, x_i} \left[ \sup_{f \in \mathcal{F}} \left| \sum_{i=1}^n \sigma_i F_f(x_i) \right| \right] + \frac{1}{n} \mathbb{E}^{\sigma_i, x_i} \left[ \sup_{f \in \mathcal{F}} \left| \sum_{i=1}^n \sigma_i (f - F_f)(x_i) \right| \right]$$

$$\le C_1 \frac{1}{\sqrt{n}} + C_2 \frac{L}{\gamma} \sqrt{\frac{c}{nd}} \underbrace{\sqrt{p \log(1 + 60 W J \tilde{\varepsilon}^{-1})}}_{\ge \sqrt{\log |\mathcal{F}_{\tilde{\varepsilon}}|}} + \frac{1}{n} \mathbb{E}^{\sigma_i, x_i} \left[ \sup_{f \in \mathcal{F}} \left| \sum_{i=1}^n \sigma_i (f - F_f)(x_i) \right| \right].$$

Here the parameter $\tilde{\varepsilon} > 0$ is related to a $\tilde{\varepsilon}$-net of $\mathcal{W}$, which we denote by $\mathcal{W}_{\tilde{\varepsilon}}$. Note, that $|\mathcal{W}_{\tilde{\varepsilon}}| \le (1 + 60 W J \tilde{\varepsilon}^{-1})^p$ (see e.g. Vershynin (2018) Corollary 4.2.13) so the same holds true for the induced net $\mathcal{F}_{\tilde{\varepsilon}} = \{\text{sgn} \circ g_w : w \in \mathcal{W}_{\tilde{\varepsilon}}\}$, which also allows us to treat the remaining expectation by subdividing the supremum:

$$\frac{1}{n} \mathbb{E}^{\sigma_i, x_i} \left[ \sup_{f \in \mathcal{F}} \left| \sum_{i=1}^n \sigma_i (f - F_f)(x_i) \right| \right] = \frac{1}{n} \mathbb{E}^{\sigma_i, x_i} \left[ \sup_{w_{\tilde{\varepsilon}} \in \mathcal{W}_{\tilde{\varepsilon}}} \sup_{w \in B_{\tilde{\varepsilon}}(w_{\tilde{\varepsilon}})} \left| \sum_{i=1}^n \sigma_i (f_w - F_{f_w})(x_i) \right| \right]$$

$$\le \frac{1}{n} \mathbb{E}^{x_i} \left[ \sup_{w_{\tilde{\varepsilon}} \in \mathcal{W}_{\tilde{\varepsilon}}} \sum_{i=1}^n |f_{w_{\tilde{\varepsilon}}} - F_{f_{w_{\tilde{\varepsilon}}}}|(x_i) \right]$$

$$+ \frac{1}{n} \mathbb{E}^{x_i} \left[ \sup_{w_{\tilde{\varepsilon}} \in \mathcal{W}_{\tilde{\varepsilon}}} \sup_{w \in B_{\tilde{\varepsilon}}(w_{\tilde{\varepsilon}})} \sum_{i=1}^n \left| |f_w - F_{f_w}|(x_i) - |f_{w_{\tilde{\varepsilon}}} - F_{f_{w_{\tilde{\varepsilon}}}}|(x_i) \right| \right]. \tag{22}$$

By Lipschitz continuity of the parameterization and of $|f - F_f|$ as derived in Equation 21, we obtain

$$\left\| |f_w - F_{f_w}| - |f_{w_{\tilde{\varepsilon}}} - F_{f_{w_{\tilde{\varepsilon}}}}| \right\|_\infty \le \frac{J}{\gamma} \tilde{\varepsilon} \quad \text{for any } w_{\tilde{\varepsilon}} \in \mathcal{W}_{\tilde{\varepsilon}} \text{ and } w \in B_{\tilde{\varepsilon}}(w_{\tilde{\varepsilon}})$$

so that

$$\frac{1}{n} \mathbb{E}^{x_i} \left[ \sup_{w_{\tilde{\varepsilon}} \in \mathcal{W}_{\tilde{\varepsilon}}} \sup_{w \in B_{\tilde{\varepsilon}}(w_{\tilde{\varepsilon}})} \sum_{i=1}^n \left| |f_w - F_{f_w}|(x_i) - |f_{w_{\tilde{\varepsilon}}} - F_{f_{w_{\tilde{\varepsilon}}}}|(x_i) \right| \right] \le \frac{J}{\gamma} \tilde{\varepsilon}.$$

Via isoperimetry and using the same bound on the cardinality of $\mathcal{F}_{\tilde{\varepsilon}}$ as before, one concludes that the first expectation in Equation 22 is of the same form as Equation 19 with subgaussian variance proxy $\sigma^2 = \frac{L^2}{\gamma^2} \frac{cn}{d}$ so that

$$\frac{1}{n} \mathbb{E}^{x_i} \left[ \sup_{w_{\tilde{\varepsilon}} \in \mathcal{W}_{\tilde{\varepsilon}}} \sum_{i=1}^n |f_{w_{\tilde{\varepsilon}}} - F_{f_{w_{\tilde{\varepsilon}}}}|(x_i) \right] = \frac{1}{n} \mathbb{E}^{x_i} \left[ \sup_{w_{\tilde{\varepsilon}} \in \mathcal{W}_{\tilde{\varepsilon}}} \sum_{i=1}^n |f_{w_{\tilde{\varepsilon}}} - F_{f_{w_{\tilde{\varepsilon}}}}|(x_i) - \mathbb{E}[|f_{w_{\tilde{\varepsilon}}} - F_{f_{w_{\tilde{\varepsilon}}}}|] \right]$$

$$+ \sup_{w_{\tilde{\varepsilon}} \in \mathcal{W}_{\tilde{\varepsilon}}} \mathbb{E}[|f_{w_{\tilde{\varepsilon}}} - F_{f_{w_{\tilde{\varepsilon}}}}|]$$

$$\le C_3 \frac{L}{\gamma} \sqrt{\frac{c}{nd}} \sqrt{p \log(1 + 60 W J \tilde{\varepsilon}^{-1})} + \sup_{w_{\tilde{\varepsilon}} \in \mathcal{W}_{\tilde{\varepsilon}}} \mathbb{E}[|f_{w_{\tilde{\varepsilon}}} - F_{f_{w_{\tilde{\varepsilon}}}}|].$$

Finally, for every $f \in \mathcal{F}$,

$$\mathbb{E}[|f - F_f|] = \int_{\mathcal{X}} |f(x) - F_f(x)| \, d\mu(x) \leq \mathbb{P}(g(x) \in [-\gamma, \gamma]). \tag{23}$$

Choosing $\gamma = \frac{S^*(g)}{2}$, we obtain by the definitions of co-margin, and once again isoperimetry (since the co-margin inherits the Lipschitzness from $g$ by design)

$$\begin{aligned}
\mathbb{P}\left(g(x) \in [-\gamma, \gamma]\right) &= \mathbb{P}\left(|g(x)| \leq \frac{S^*(g)}{2}\right) \\
&\leq \mathbb{P}\left(|h_g^*(x) - S^*(g)| \geq \frac{S^*(g)}{2}\right) \\
&\leq 2 \exp\left(-\frac{d \, S^*(g)^2}{8cL(g)^2}\right) \leq 2 \exp\left(-\frac{d \, S^{*2}}{8cL^2}\right) = 2 \exp\left(-\frac{d \, \bar{S}^{*2}}{8c}\right).
\end{aligned}$$

Putting it all together, we have

$$\mathcal{R}_{n,\mu}(\mathcal{F}) \leq C_1 \frac{1}{\sqrt{n}} + C_2' \frac{L}{S^*} \sqrt{\frac{c}{nd}} \sqrt{p \log(1 + 60 W J \tilde{\varepsilon}^{-1})} + \frac{2J}{S^*} \tilde{\varepsilon} + 2 \exp\left(-\frac{d \, S^{*2}}{8cL^2}\right).$$

$\square$

## F  MULTI-CLASS CLASSIFICATION

In this section, we briefly outline how our results extend to categorical distributions with $\mathcal{C} \in \mathbb{N}$ classes. We assume that a classifier is given by

$$f : \mathcal{X} \to \{0, 1\}^{\mathcal{C}},$$

with exactly one non-zero entry for each $x \in \mathcal{X}$. The additional regularity assumption (H3)$'$, the adaptations of the conditions in (H3) to the multi-class setting can be formalized as follows.

(H3)'  The hypothesis class has the form $\mathcal{F} = \operatorname{argmax} \circ \mathcal{G}$, where $\mathcal{G} = \{g_w : \mathcal{X} \to [0, 1]^{\mathcal{C}} : w \in \mathcal{W}\}$ is a parameterized family of Lipschitz functions. The parameter space $\mathcal{W} \subset \mathbb{R}^p$ is bounded with $\operatorname{diam}(\mathcal{W}) \leq W$, and the parameterization is Lipschitz:

$$\|g_{w_1} - g_{w_2}\|_\infty \leq J \|w_1 - w_2\|.$$

Thus, we can interpret $g \in \mathcal{G}$ as representing the class probabilities.

**Remark 20.** *For binary classification, i.e. $\mathcal{C} = 2$, the classifiers are of the form $f : \mathcal{X} \to \{0, 1\}^2$, instead of $f : \mathcal{X} \to \{-1, 1\}$, as considered earlier. However, one can translate between these representations by post-composing with either*

$$\alpha(x_1, x_2) := x_1 - x_2 \quad \text{or} \quad \beta(x) := \left(\frac{x+1}{2}, \frac{1-x}{2}\right).$$

*By the contraction principle for Rademacher complexity, it is therefore sufficient to compute the complexity for one of these models.*

As in the binary case, our proofs start by considering the Rademacher complexity of the function class $\mathcal{F}$:

$$\mathcal{R}_{n,\mu}(\mathcal{F}) = \frac{1}{n} \mathbb{E}^{\sigma_{ij}, x_i} \left[ \sup_{f \in \mathcal{F}} \Big| \sum_{i=1}^n \sum_{j=1}^{\mathcal{C}} \sigma_{ij} f_j(x_i) \Big| \right] \leq \sum_{j=1}^{\mathcal{C}} \frac{1}{n} \mathbb{E}^{\sigma_{ij}, x_i} \left[ \sup_{f \in \mathcal{F}} \Big| \sum_{i=1}^n \sigma_{ij} f_j(x_i) \Big| \right].$$

Each summand corresponds to a binary classification problem with a one-vs-all classifier $f_j$. Indeed, $f_j$ is $\frac{2}{S(f)-t}$-Lipschitz on $A_t(f)$. Transforming via

$$f_j \mapsto 2f_j - 1 : \mathcal{X} \to \{-1, 1\},$$

we can follow the same reasoning as in Appendix C, obtaining, up to a linear factor of $\mathcal{C}$, the same result as the first part of Theorem 4, generalized to the multi-class setting.

Table 1: Multi-class definitions.

| Concept | Definition |
|---|---|
| Isoperimetry | $\mathbb{P}(\|f(x) - \mathbb{E}[f]\|_\infty \geq t) \leq 2\exp\left(-\frac{dt^2}{2cL^2}\right)$ |
| Rademacher complexity | $\mathcal{R}_{n,\mu}(\mathcal{F}) = \frac{1}{n}\mathbb{E}^{\sigma_{i,j},x_i}\left[\sup_{f\in\mathcal{F}}\left|\sum_{i=1}^n \sum_{j=1}^{\mathcal{C}} \sigma_{ij} f_j(x_i)\right|\right]$ |
| Margin | $h_f(x) = \sum_{j=1}^{\mathcal{C}} h_f^j(x), \quad h_f^j(x) := \inf\{\|x - z\|_2 : f(z) \neq j,\, z \in \mathbb{R}^d\}$ |
| Class stability | $S(f) = \sum_{j=1}^{\mathcal{C}} S(f)^j, \quad S(f)^j := \mathbb{E}[h_f^j]$ |
| Co-margin | $h_g^*(x) = \sum_{j=1}^{\mathcal{C}} h_g^{*j}(x), \quad h_g^{*j}(x) := \max\left(0, g_j(x) - \max_{i\neq j} g_i(x)\right)$ |
| Co-stability | $S^*(g) = \sum_{j=1}^{\mathcal{C}} S^{*j}(g), \quad S^{*j}(g) := \mathbb{E}[h_g^{*j}]$ |

Similarly, under assumption (H3), we can write

$$2f_j - 1 = \text{sgn}\left(g_j - \max_{i\neq j} g_i(x)\right),$$

which allows us to proceed as in Appendix E to obtain a multi-class generalization of the second part of Theorem 4 and Theorem 13. The only minor difference lies in bounding the term in Equation 23:

$$\mathbb{E}[|f_j - F_{f_j}|] \leq \mathbb{P}\left[|g_j(x) - \max_{i\neq j} g_i(x)| \leq \gamma\right].$$

Choosing $\gamma = \frac{S^*(g)}{2}$, we use that for all $j$, $|g_j(x) - \max_{i\neq j} g_i(x)| > h_g^*(x)$, which yields

$$\mathbb{P}\left[|g_j(x) - \max_{i\neq j} g_i(x)| \leq \frac{S^*(g)}{2}\right] \leq \mathbb{P}\left[|h_g^*(x) - S^*(f)| \geq \frac{S^*(g)}{2}\right]$$

$$\leq 2\exp\left(-\frac{d\,S^*(g)^2}{8cL(g)^2}\right)$$

$$\leq 2\exp\left(-\frac{d\,S^{*2}}{8cL^2}\right) = 2\exp\left(-\frac{d\,\bar{S}^{*2}}{8c}\right).$$

We conclude that all of our results extend to the multi-class case. Moreover, the measure used in our MNIST and CIFAR-10 experiments (Section 6) is the correct generalization.

## G  EXPERIMENTAL DETAILS FOR STABILITY MEASUREMENT

**Training setup.**  We trained fully connected MLPs with 4 or 8 hidden layers on MNIST and CIFAR-10, using widths $w \in \{128, 256, 512, 1024, 2048\}$. For CIFAR-10, we additionally evaluated convolutional neural networks (CNNs) with widths $w \in \{128, 256, 512, 1024\}$. On MNIST, we considered Heaviside-activation MLPs to examine whether the observed scaling behaviour extends to discontinuous score functions.

All MLP models use ReLU activations (or Heaviside activations where specified) and batch normalization. Optimization was performed with Adam (Kingma & Ba, 2015) for embedding and output layers and Muon (Jordan et al., 2024) for hidden layers, using a batch size of 256 and learning rate $10^{-3}$. Training continued until at least 99% training accuracy was achieved, yielding near-interpolating solutions.

**Additional MNIST protocols.**  For 8-layer MLPs on MNIST, we additionally performed two controlled variations: (i) extended-width experiments with $w$ up to 16384, and (ii) fixed-epoch training runs in which all widths were trained for seven epochs, independent of when 99% training accuracy was first reached (all models attained 99% within this budget). These experiments isolate the effect of overparameterization from heterogeneous training durations.

**Parameter counts and normalization.** For each model, we recorded the total number of trainable parameters $p$, input dimension $d$, and total number of training samples $n$.

**Stability estimation.** Class stability $S(f)$ is estimated via $\ell_2$ adversarial perturbations using Foolbox (Rauber et al., 2017). We apply a collection of $\ell_2$ attacks (including Deep-Fool and $\ell_2$-PGD (Moosavi-Dezfooli et al., 2016; Madry et al., 2018)) over an $\varepsilon$-grid $\{0.01, 0.05, 0.1, 0.2, 0.5, 1.0, 2.0\}$. For each input $x$, we record the minimum achieved perturbation norm $\min \|x_{\mathrm{adv}} - x\|_2$ across all attacks and $\varepsilon$ values. If no attack succeeds within the maximum $\varepsilon$, we set the perturbation norm to $\max(\varepsilon)$. For the chosen $\varepsilon$ grid, attack success rates exceed 90% across model sizes, and are typically higher for smaller widths. Failures occur primarily for the largest models, where no adversarial example is found within the maximal radius. In these cases, the assigned value $\max(\epsilon)$ induces a downward bias, so the reported stability values are conservative lower bounds.

**Normalized Co-Stability estimation.** The empirical co-stability $S^*(g)$ is computed via the multi-class margin

$$g_j(x) - \max_{i \neq j} g_i(x), \qquad j = \arg\max_i g_i(x),$$

averaged over the dataset. We estimate the Lipschitz constant $L(g)$ using the efficient ECLIPSE method (Xu & Sivaranjani, 2024), and report the normalized ratio $S^*(g)/L(g)$ as a function of model size.

**Implementation.** Training and evaluation code is implemented in PyTorch (Paszke et al., 2019). For MLPs, images were flattened to vectors. Attack evaluations were conducted over the test dataset.

**Reproducibility.** All experiments were run with multiple random seeds $\{0, 1, 2, 3, 4\}$, and mean with standard deviation are reported.

# H    ADDITIONAL EXPERIMENTS

We report additional experiments for MLPs trained on MNIST and CIFAR-10 with varying depths. Unless stated otherwise, MLP widths are $w \in \{128, 256, 512, 1024, 2048\}$. Each figure shows class stability (left) and normalized co-stability (right) as a function of width.

## H.1    CIFAR 10

Figure 5 illustrates the width-dependent scaling of 4-layer MLPs on CIFAR-10. The observed increase in stability aligns with the qualitative trend indicated by our theoretical analysis and is comparable to the behaviour of the 8-layer models.

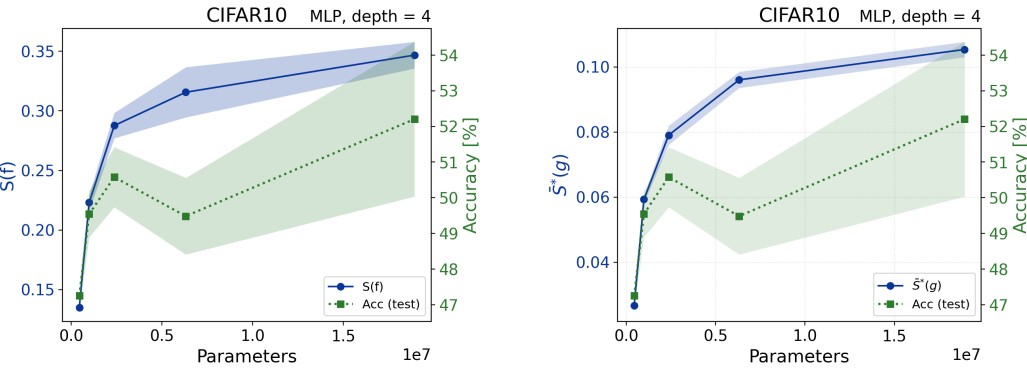

Figure 5: 4-layer MLPs on CIFAR10.

## H.2   MNIST

Figure 6 indicates a monotonic increase in stability with width for 4-layer MLPs, in line with the qualitative behaviour suggested by our theory. In contrast, the 8-layer models in Figure 7 display an initial decrease in stability for small widths.

We attribute this effect to heterogeneous training durations. To reach 99% training accuracy, narrow models required up to seven epochs, whereas the widest models required only two, with a monotonic trend across widths. Since cross-entropy training continues to increase margins even after achieving correct/perfect classification, training duration, especially in the low epoch regime, induces systematically smaller stability values.

We test this hypothesis in two ways. First, we extend the width range up to $w = 16384$ (Figure 8). For sufficiently large widths (with inherently similar/equal number of epochs), the predicted monotonic relation between model size and stability is recovered. Second, we fix the training duration to seven epochs for all widths (Figure 9). Under uniform training, the expected scaling behaviour is restored across the full width range.

The same effects are observed for normalized co-stability.

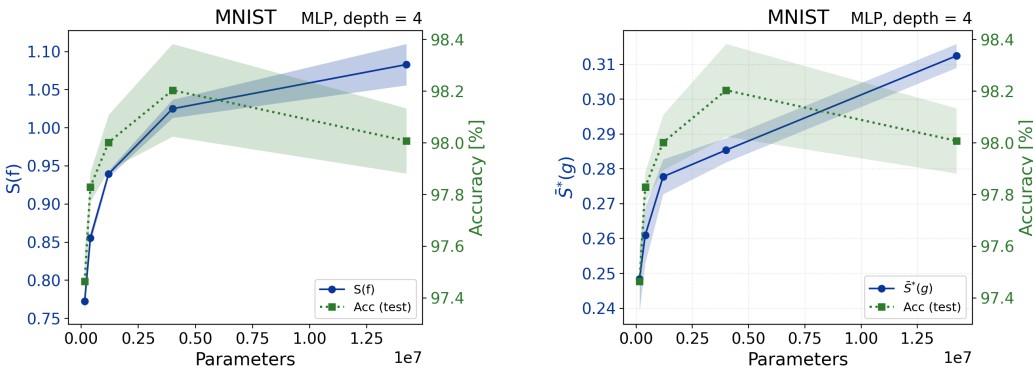

Figure 6: 4-layer MLPs on MNIST.

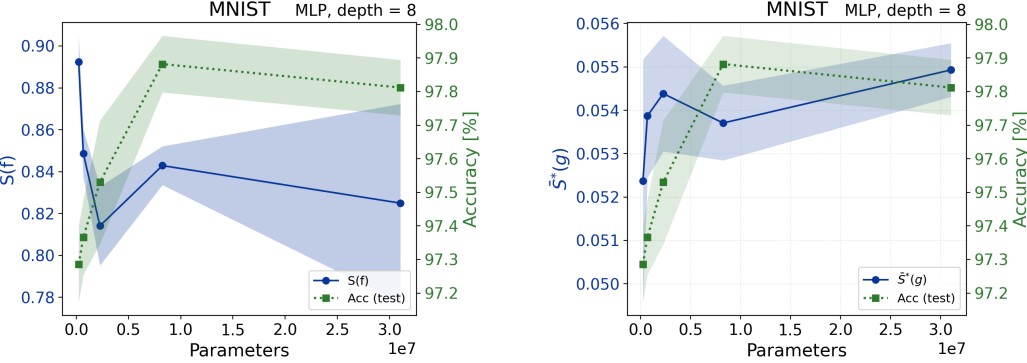

Figure 7: 8-layer MLPs on MNIST.

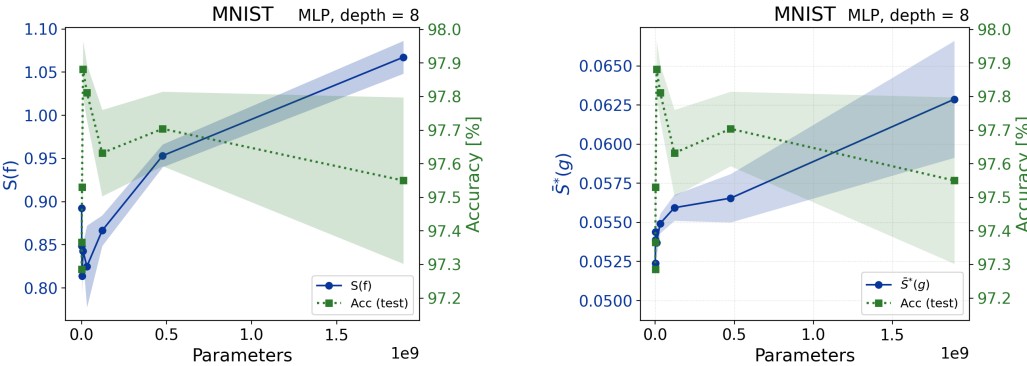

Figure 8: 8-layer MLPs on MNIST with extended widths up to $w = 16384$.

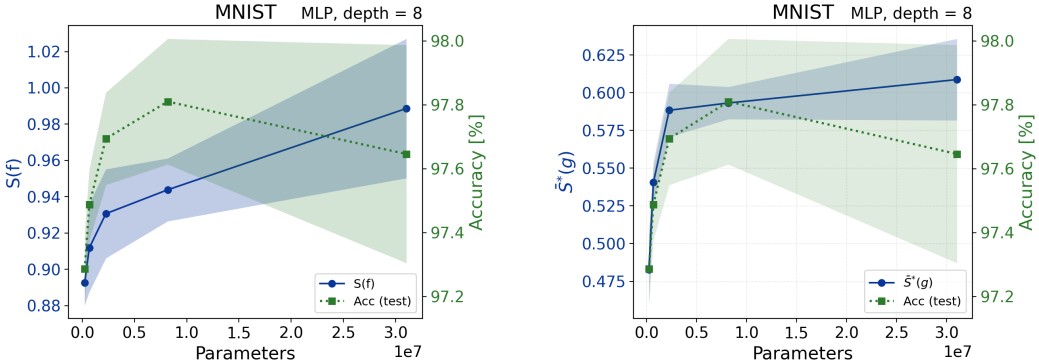

Figure 9: 8-layer MLPs on MNIST trained for 7 epochs.

