# OpenReview forum: "The Price of Robustness:  Stable Classifiers Need Overparameterization"
_ICLR.cc/2026/Conference — ICLR 2026 Poster_

### Official Review · Reviewer_qsC1 · 2025-10-18

**Soundness:** 2
**Presentation:** 3
**Contribution:** 2
**Rating:** 6
**Confidence:** 2

**Summary:**

The paper defines two robustness/stability surrogates for discontinuous classifiers and proves data‑dependent generalization bounds that tighten as stability increases: class stability $S(f)$ and normalized co‑stability $\bar{S}^*$. Under a c-isoperimetry assumption on the data distribution (Def. 3, p. 4), the authors prove a Rademacher complexity bound for finite classifier classes, and an extension to infinite, parameterized classes via normalized co‑stability and parameter‑Lipschitz score maps. Experiments are conducted On MNIST and CIFAR‑10 with fully‑connected MLPs.

**Strengths:**

- Framing of robustness for discontinuous classifiers. Replacing Lipschitzness of $f$ with expected input‑margin and codomain margin is sensible and bridges a known gap in extending the Bubeck–Sellke robustness law to classification. The formalization via signed‑distance representation is clean.
- The finite‑class bound (Theorem 4) uses a careful Lipschitz surrogate + isoperimetry argument, then sharpens by invoking the signed‑distance representation. The infinite‑class extension cleanly separates the roles of average confidence and smoothness.

**Weaknesses:**

- Assumptions vs. practice gap. The isoperimetry requirement on $\mu_X$ is strong and not stress‑tested. The paper states a manifold‑dimension interpretation, but no empirical probes of isoperimetry or concentration are provided, even in toy data. The external validity of the law depends on this.
- Overparameterization claim feels over‑indexed to $nd$. The corollaries argue a necessity of $p≈nd$, but the experiments do not stress this scaling (e.g., sweeping $n$ and $d$ while reading off the stability needed), nor do they test architectures beyond MLPs. As written, the claim risks overreach.

**Questions:**

- Assumption stress‑test. Can you empirically probe the isoperimetry assumption (e.g., concentration of Lipschitz functions) on MNIST/CIFAR embeddings, or supply a synthetic non‑isoperimetric counterexample where your bound degrades?
- Is the theoretical claim / empirical phenomena general enough, to hold on other classifiers, such as SVM, random forest classifiers, etc (other than MLP)? Will one obtain similar experiment results with different network architectures?
- typos: “Selke” should be “Sellke”

---

> ### Author Response · Authors · 2025-11-21
>
> We thank the reviewer for their valuable feedback on the empirical perspective on our results. We will directly answer the questions and also provide our comments on the mentioned weaknesses.
>
> We also thank the reviewer for pointing out the typo, which is corrected throughout the paper.
>
> **Q: theoretical claim / empirical phenomena general enough**
> In Appendix F, we show that the binary setting $\mathrm{sgn}\circ g$ extends naturally to the multiclass case by reducing $\mathrm{argmax}\circ g$ to multiple one vs all problems. Consequently, SVMs and essentially all modern deep-learning classifiers fit our hypothesis class (assuming the activation function is Lipschitz, as is standard): they are of the form $\mathrm{argmax}\circ g$ with $g$ Lipschitz continuous. In particular, MLPs, CNNs, and RNNs are Lipschitz architectures in both input space and parameter space. For self-attention, the Lipschitz constant can be comparatively large, which is an active area of research (see, e.g., [1]).
>
> It is therefore difficult to identify practically relevant counterexamples that do not satisfy our structural assumptions. The most direct counterexamples are models with inherently non-Lipschitz operations, such as Heaviside networks or architectures employing discontinuous activation functions. Classical non-neural models such as random forests also fall outside this framework, since their decision functions are piecewise-constant with an unbounded Lipschitz constant. However, these architectures are still covered by Theorem 4.1.
>
> We also temporarily added numerical experiments (please see Appendix H) with CNNs and Heaviside ANNs, showing similar behaviour as MLPs and thus supporting our theory. Experiments with ViTs are currently running and will be added to Appendix H once terminated.
>
> [1] Laker Newhouse, R. Preston Hess, Franz Cesista, Andrii Zahorodnii, Jeremy Bernstein, and Phillip Isola. Training Transformers with Enforced Lipschitz Constants. arXiv:2507.13338. 2025

---

> ### Author Response · Authors · 2025-11-21
>
> **Q: Assumption stress‑test.**
> We consider our work to be mainly theoretical, and the experiments only aim to support the general claim of a relationship between overparameterization and stability. Exhaustively validating the isoperimetry assumption is a hard problem beyond the scope of this work; however, we next describe our approaches to provide some hints/intuitions on the empirical reasonability of the assumption.
>
> Since we can only evaluate finitely many functions on finitely many datapoints, any empirical distribution satisfies the isoperimetry condition for a sufficiently large constant. The relevant question is therefore whether one can obtain a reasonably small and reliable effective isoperimetric constant for MNIST and CIFAR-10. We conducted the following experiments, with detailed results provided in the updated Appendix H.
>
> *Experimental protocol.*
> For several families of functions $f$---random normalized linear functions, $1$-Lipschitz constrained MLPs, unconstrained random MLPs, and the trained classifier margin functions---we computed the empirical distribution of
> $|f(x_i)-\mathbb{E}[f]|$ over the full dataset. For a grid of radii $t$, we estimated the tail probabilities $P(|f-\mathbb{E}f|>t)$ and performed a linear regression of $\log P(|f-\mathbb{E}f|>t)$ against $t^2$. This corresponds to estimating the sub-Gaussian/isoperimetric form of the tail. We report the regression quality (R$^2$) and the inferred effective constant $\hat{c}$ for each function class.
>
> *Validation on Gaussian toy data.*
> We generated several synthetic datasets in $\mathbb{R}^{784}$, constructed a $10$-class linearly separable task, and used Gaussians with varying variance $\sigma^2$ (with true isoperimetric constant $c=d\sigma^2$). Linear functions and $1$-Lipschitz networks recovered constants of the correct order. Random MLPs, however, yielded a deviation up to $10^{16}$ from the correct order, showing that worst-case Lipschitz function classes did not recover the actual isoperimetry constants in the experiments. Trained networks produced intermediate values: significantly smaller than worst-case MLPs, but moderately larger than the linear/Lipschitz probes. Thus, we conclude that linear functions, $1$-Lipschitz networks, and, to a certain degree, trained networks recover the underlying constant, whereas random MLPs were off by a huge margin and should also not be considered reliable for MNIST/CIFAR-10.
>
> *Results on MNIST and CIFAR-10.*
> Applying the same procedure to MNIST and CIFAR-10, we observed:
> - Random $1$-Lipschitz linear functions and structured $1$-Lipschitz networks exhibit tight quadratic log-tails, with $\hat{c} \approx 10$-$50$ for MNIST and  $30$-$180$ for CIFAR-10, and median R$^2\ge 0.99$.
> - Random MLPs require much larger constants (median $\hat{c} \sim 10^{12}$ on MNIST and $10^{14}$ on CIFAR-10).
> - For the margin functions
> 	$$ 	f(x)=\ell_y(x)-\max_{k\neq y}\ell_k(x)$$
> 	of trained classifiers, we obtain R$^2\approx 0.94$-$0.98$ and   $\hat{c} \approx 10^2$-$10^3$ on MNIST and CIFAR-10. These values are much smaller than those for worst-case MLPs and only moderately larger than for linear/$1$-Lipschitz probes. This suggests that the data distribution behaves "near-isoperimetrically'' for the practically relevant function classes underpinning our theory.
>
>
> *Synthetic non-isoperimetric stress test.*
> For unbounded heavy-tailed distributions (e.g., Cauchy), the margin integrals defining stability are not finite. Instead of such pathological cases, we tested Gaussians with increasing variance to simulate weaker concentration. Using the same linearly separable toy task, we trained MLPs of various widths and observed that the stability-accuracy correlation deteriorates with growing variance; for $\sigma^2=0.1$ it disappears entirely, with stability even decreasing with width. This indicates that concentration is needed to reproduce the MNIST/CIFAR-10 behavior. Although MNIST and CIFAR-10 yield empirical constants similar to the $\sigma^2=0.1$ toy case (where the stability–width relationship degrades), the theory can be relaxed to mixtures of isoperimetric distributions. In this setting the relevant constant is the maximum over the modes, while the global mixture constant may be much larger. This suggests that the true per-mode constants in MNIST/CIFAR-10 are smaller than our global empirical estimates.
>
> *Summary.*
> Our estimator correctly identifies isoperimetry when ground truth is known, and MNIST/CIFAR-10 show isoperimetric behavior with moderate constants for relevant function classes (linear probes, structured Lipschitz networks, and trained margins), whereas worst-case Lipschitz networks and weakly concentrated data require much larger constants. This supports the practical plausibility of our assumption. The theory extends to mixtures of concentrated components, and also weaker forms of concentration for the relevant function classes (e.g., trained MLP margins) are sufficient to recover the theory.

---

> > ### Comment · Reviewer_qsC1 · 2025-11-25
> >
> > Thank you for the response. I will keep my score.

---

### Official Review · Reviewer_WFc2 · 2025-11-01

**Soundness:** 3
**Presentation:** 2
**Contribution:** 1
**Rating:** 4
**Confidence:** 3

**Summary:**

This paper extends the "universal law of robustness," which analyzes the relationship between overparameterization, stability, and generalization, to the domain of discontinuous classifiers. The authors prove that for any discontinuous classifier, overparametrization is necessary if one wants to robustly interpolate the data. The prior work by Bubeck and Selke (2021) established this law for high-dimensional Lipschitz functions, but their reliance on the Lipschitz constant is problematic for discontinuous classifiers. To address this, the authors leverage alternative stability measures: the input-space margin concept of "class stability" and a newly introduced output-space metric called "normalized co-stability." By employing these two measures, they successfully extend the theoretical framework to the discontinuous case, providing a compelling explanation for why modern, heavily overparameterized models can achieve robust generalization.

**Strengths:**

A major strength of this paper lies in its successful extension of the theoretical framework beyond the original Lipschitz assumption to the more challenging domain of discontinuous classifiers. While the work is theoretical in nature, its findings have practical implications explaining why heavily overparameterized models, which are common in modern machine learning, can achieve robust generalization. Furthermore, the paper is well-written and organized.

**Weaknesses:**

Its novelty feels somewhat incremental. At a high level, the core idea mirrors that of Bubeck and Selke (2021): the original work used the Lipschitz constant to ensure that for different inputs $x_i$ and $x_j$, the distance $\|| x_i - x_j\||$ is non-trivial (i.e., $\Omega(1)$); this paper adopts a similar underlying principle.

For finite function classes, the authors use the "class stability" proposed by Liu and Hansen (2024) to derive an upper bound on the Rademacher complexity. However, as noted in Remark 5, this bound introduces an undesirable additional factor in certain regimes, which requires further assumptions to mitigate.

For the infinite function class case, the analysis is restricted to a specific function class of the form $\mathrm{sgn}\circ \mathcal{G}$, where $\mathcal{G}$ is Lipschitz. This raises questions about how broadly the results can be generalized to all discontinuous classifiers. The newly proposed "co-stability" measure, used to derive Theorem 13 and Corollary 15, is defined in terms of this Lipschitz constant of $\mathcal{G}$. Consequently, the results feel analogous to the original idea by Bubeck and Selke, with the main difference being the consideration of a composed function class $\mathrm{sgn}\circ\mathcal{G}$ rather than a fundamentally new approach.​​

Typo:
page 4: The first bullet in thm4: empirical Rademacher complexity -> Rademacher complexity

**Questions:**

Q1.  About $\mathrm{sgn}\circ \mathcal{G}$. While it clearly covers classical models like SVMs, its application to modern deep learning architectures merits further clarification. Could you provide examples of overparameterized deep learning models that fit the definition of a discontinuous classifier in this paper, and perhaps more importantly, any that might not fit this structure?

Q2. The original Bubeck & Selke (2021) paper required a lower bound on the data dimension $d$ relative to the level of robustness $\varepsilon$ (Assumption 4 in their paper). This assumption appears to be absent in your work. Could you highlight why this dimensional lower bound is no longer necessary in your proof for discontinuous classifiers?

Q3. The theoretical bounds you derive depend on the true data distribution, which is unknown in practice, raising the possibility that the bounds could be vacuous. While the experiments effectively show a correlation between the proposed stability measures and test performance, could you compute the actual upper bounds derived in the paper using empirical estimates? How do these computed bound values compare to the observed generalization error?

---

> ### Author Response · Authors · 2025-11-21
>
> We thank the reviewer for their thoughtful feedback. Below we offer our perspective on the mentioned weaknesses and then address the specific questions.
>
>  *"Its novelty feels somewhat incremental. [...]"*:
> We agree that both our work and Bubeck&Sellke (2021) aim to relate an appropriate notion of robustness to model complexity and overparameterization. Our stability measure can indeed be viewed as a classifier-specific analogue of Lipschitz regularity: while the global Lipschitz constant is infinite for discontinuous classifiers, stability quantifies a data-dependent average distance to the decision boundary and thus plays a similar role to an average local Lipschitz control. The difference is therefore not in the underlying philosophy, but in the fact that the global  'sup' notion used in the continuous setting is simply unavailable in classification.
>
> Importantly, stability thus imposes much weaker geometric constraints than Lipschitz continuity, and it is a priori unclear whether such average control is sufficient to recover any robustness--overparameterization tradeoff. Our main contribution is to show that this classification-tailored regularity (class stability and its output-space analogue, co-stability) can indeed serve as an appropriate replacement for Lipschitzness and still yields an overparameterization requirement of the same order in the discontinuous setting.
>
> We also note that classification is the natural setting for adversarial robustness, where discontinuous decision boundaries and margin-based notions of stability are central. Extending the Bubeck&Sellke theory to this setting is therefore not only mathematically nontrivial but also practically important.
>
> *"For finite function classes, the authors use the "class stability" proposed by Liu and Hansen (2024) to derive an upper bound on the Rademacher complexity. [...]"*:
> We acknowledge that Theorem 4.1 introduces an additional factor in the Rademacher complexity bound. This factor arises because Theorem 4.1 makes no regularity assumptions on the class of discontinuous classifiers. Without any assumptions, the hypothesis class can contain highly irregular or even pathological functions (e.g., the Dirichlet function), and the bound must reflect this worst-case behavior.
>
> Note that the same factor appears in the bound of Bubeck-Sellke (under Lipschitz regularity) if one assumes only a Poincaré inequality on the measure instead of isoperimetry (see p.4 in their paper). Thus, there seems to be a trade-off between assumptions on the data and model regularity.
>
> For practical interpretation, the relevant result is Theorem 4.2, which removes this extra factor. The assumptions required here -- path-connectedness of the input space and closedness of the positive-class preimage -- are extremely mild and satisfied by essentially all classifiers encountered in practice. In fact, for any binary classifier of the form $\mathrm{sgn}\circ g$ (with either convention $\mathrm{sgn}(0)=1$ or $\mathrm{sgn}(0)=0$), these assumptions are automatically met; the equivalence of these two characterizations is established in Lemma 18 in the Appendix.
>
> We also refer the reviewer to our response to Q1 below, where we discuss the generality of composed function classes of the form $\mathrm{sgn}\circ g$ in modern deep learning models.
>
> *"Consequently, the results feel analogous to the original idea by Bubeck and Sellke [...]"*:
> Our co-stability analysis yields implications that go beyond the results of Bubeck&Sellke. Their work shows that controlling the Lipschitz constant of a real-valued predictor can improve generalization. For classification, however, this is insufficient: our results show that one must control both the smoothness of $g$ (for classifiers of the form $\mathrm{sgn}\circ g$) and the co-domain margin (average confidence) to guarantee good generalization despite overparameterization.

---

> ### Author Response · Authors · 2025-11-21
>
> *"[...] the main difference being the consideration of a composed function class $\mathrm{sgn}\circ \mathcal{G}$ rather than a fundamentally new approach.​​"*:
> Although our infinite-class analysis is formulated in terms of composed classes of the form $\mathrm{sgn}\circ \mathcal{G}$, extending bounds from $\mathcal{G}$ to $\mathrm{sgn}\circ \mathcal{G}$ is not straightforward. For regular transformations $f$ (e.g., $f$ Lipschitz), one can often relate $\mathcal{R}(f\circ \mathcal{G})$ to $\mathcal{R}(\mathcal{G})$, where $\mathcal{R}$ denotes the Rademacher complexity of the function class. For discontinuous transformations such as $\mathrm{sgn}$, however, such reductions typically fail: even a single discontinuity can substantially increase the combinatorial complexity of the resulting classifier class.
>
> A concrete illustration is provided by the class $\mathrm{MLP}_1$ of $1$-Lipschitz ReLU networks. By simple rescaling, the classifier classes $\mathrm{sgn}\circ \mathrm{MLP}_1$ and $\mathrm{sgn}\circ \mathrm{MLP}$ coincide. Yet the VC dimension of $\mathrm{sgn}\circ \mathrm{MLP}$ is known to be $\mathrm{VCdim}(\mathrm{sgn}\circ \mathrm{MLP}) = \Theta(WL\log W)$, where $W$ denotes the number of weights and $L$ the depth [1]. Hence, imposing a global Lipschitz constraint on the score function does not reduce the combinatorial complexity of the induced classifier.
>
> In order to extend from the finite to the infinite-class setting, we require the explicit representation $\mathrm{sgn}\circ \mathcal{G}$ mainly to ensure that $g\in  \mathcal{G}$ is Lipschitz in its parameters, which allows us to control covering numbers for the function class. This is a standard technical requirement for infinite hypothesis classes and is unrelated to the input-space Lipschitzness used in Bubeck&Sellke.
>
> Finally, we again refer the reviewer to our response to Q1, where we motivate why virtually all classifiers in practice satisfy this structural representation.
>
> [1] Bartlett, Harvey, Liaw, and Mehrabian, Nearly-tight VC-dimension and Pseudodimension Bounds for Piecewise Linear Neural Networks. JMLR 2019.
>
>
>
>
> We also thank the reviewer for pointing out the typo on p.4, which we corrected accordingly.

---

> ### Author Response · Authors · 2025-11-21
>
> Next, we answer the questions Q1-Q3 posed by the reviewer.
>
> **Q1**:
> In Appendix F, we show that the binary setting $\mathrm{sgn}\circ g$ extends naturally to the multiclass case by reducing $\mathrm{argmax}\circ g$ to multiple one vs all problems. Consequently, essentially all modern deep-learning classifiers fit our hypothesis class (assuming the activation function is Lipschitz, as is standard): they are of the form $\mathrm{argmax}\circ g$ with $g$ Lipschitz continuous. In particular, MLPs, CNNs, and RNNs are Lipschitz architectures in both input space and parameter space. For self-attention, the Lipschitz constant can be comparatively large, which is an active area of research (see, e.g., [1]).
>
> It is therefore difficult to identify practically relevant counterexamples that do not satisfy our structural assumptions. The most direct counterexamples are models with inherently non-Lipschitz operations, such as Heaviside networks or more generally architectures employing discontinuous activation functions. Classical non-neural models such as random forests also fall outside this framework, since their decision functions are piecewise-constant with an unbounded Lipschitz constant. However, these architectures are still covered by Theorem 4.1.
>
> We also temporarily added numerical experiments (please see Appendix H) with CNNs and Heaviside ANNs, showing similar behaviour as MLPs and thus supporting our theory. Experiments with ViTs are currently running and will be added to Appendix H once terminated.
>
> [1] Laker Newhouse, R. Preston Hess, Franz Cesista, Andrii Zahorodnii, Jeremy Bernstein, and Phillip Isola. Training Transformers with Enforced Lipschitz Constants. arXiv:2507.13338. 2025
>
> **Q2**:
> Assumption 4 is not a necessary condition to derive the law of robustness result in their framework. They provide two versions of their result, one with and one without Assumption 4. Via Assumption 4, the lower bound on the Lipschitz constant is improved by a factor $\sigma$, where $\sigma^2$ denotes the noise level (see Assumption 3 in Theorem 4 in their paper and the paragraph below Theorem 4).
>
> Similarly, we could add this mild assumption on the input dimension and improve the constant in our asymptotic result.
>
> **Q3**:
> We computed the constants in our generalization bound and obtained $$\mathcal{R}_{n,\mu}(\mathcal{F}) \leq \frac{1}{\sqrt{n}} + 12\sqrt{2} \frac{1}{S}\sqrt{\frac{c \log |\mathcal{F}|}{n d}} + 2\exp\left(-\frac{d S^{2}}{8c}\right).$$
>
> In particular, this bound can be rewritten as a condition on the minimal stability $S$ required for the bound to be non-vacuous:
> $$ S \geq \sqrt{\frac{288 c \log|\mathcal{F}|}{n d}}. $$
>
> As discussed in the paper, we typically have $\log|\mathcal{F}| \approx p$, with $p$ being the number of parameters. For instance, for MLPs with FP32 weights and biases, we obtain the crude upper bound $ \log|\mathcal{F}| \leq  32 p \log 2$.
> Thus, the only unknown quantity in the bound is the isoperimetric constant $c$.
>
> In our response to Reviewer qsC1, we highlight the difficulties of empirically validating the isoperimetry assumption and describe an experimental procedure for estimating an empirical proxy $\hat{c}$ of the isoperimetry constant $c$ on MNIST and CIFAR-10. Experiments on synthetic Gaussian data show that $\hat{c}$ correlates well with the true constant, although its absolute value cannot be interpreted directly and depends strongly on the function class being tested. We refer the reviewer to Appendix H for the quantitative results. Nevertheless, the numerical constants in the bound are small, so whether the bound is non-vacuous depends on the interplay among $p$, $c$, $n$, and $d$.
>
> As an illustrative example, in a high-dimensional overparameterized regime with  $p \approx \frac{n\sqrt{d}}{10}$ and $c = \Omega(1)$, the condition above would require only $S \ge 1$ for the bound to be non-vacuous. Our empirical results suggest that $S \approx 1$ is indeed a realistic stability value for classifiers on MNIST and CIFAR-10.

---

> > ### Comment · Reviewer_WFc2 · 2025-11-27
> >
> > Thank you for your detailed response to my initial question. I have understood most of it; however, I have two follow-up questions regarding specific details.
> >
> > **(1) Our results show that one must control both the smoothness of  (for classifiers of the form ) and the co-domain margin (average confidence) to guarantee good generalization despite overparameterization.**
> >
> > Since the assumption (H3) enforces the image of $g$ to be a subset of $[-1,1]$, $ {S}^{\*}(g) $  is always bounded by 1. Even though it is an expectation, it seems to be a constant independent of $n$,$p$, and $d$. Consequently, in the context of overparameterization, it seems that the results involving $ {S}^{\*}(g) $ are actually discussing $L(g)$. Could you clarify a specific role of $ {S}^{\*}(g) $  in this context?
> >
> >
> >
> > **(2) This is a standard technical requirement for infinite hypothesis classes and is unrelated to the input-space Lipschitzness used in Bubeck&Sellke.**
> >
> > I was confused by the comment that your technical requirement is unrelated to the input-space Lipschitzness used in Bubeck&Sellke.
> > As far as I know, both Bubeck & Sellke’s work and this study assume Lipschitz continuity with respect to the parameters (Bubeck&Sellke, Theorem 3 - Assumption 1) to characterize the optimal bound of the Lipschitz constant of
> > $f$ with respect to the input space. Thus, I couldn't understand your statement that it is "unrelated." Could you help me understand what makes your approach different?

---

> > > ### Author Response · Authors · 2025-11-27
> > >
> > > We are glad our earlier response was helpful and address the reviewer’s follow-up questions below.
> > >
> > > **(1):**
> > > As the reviewer correctly notes, the co-stability $S^{\ast} (g)$ is uniformly upper bounded by 1. However, its role in the overparameterization setting arises through the key relation (Equation 8),
> > > $$\frac{S^{*}(g)}{L(g)} \le C \sqrt{\frac{p}{nd}}\quad\Longrightarrow\quad
> > > L(g) >  C^{-1} S^{\ast}(g)\sqrt{\frac{nd}{p}},
> > > $$
> > > where $C>0$ is independent of $n$, $p$, and $d$. Thus, obtaining a lower bound on $L(g)$ necessarily requires a _lower bound_ on $S^{\ast}(g)$. The upper bound $S^{\ast}(g)\le 1$ only reflects the best-case scenario of maximal co-stability $S^{\ast}(g) =  1$; if $S^{\ast}(g)$ is small, the bound in Equation 8 becomes uninformative, and we cannot conclude anything meaningful about $L(g)$.
> > >
> > > This shows that $S^{\ast}(g)$ – the average co-margin – is essential for deriving our robustness–overparameterization tradeoff for infinite function classes. Its presence is not an artifact but a consequence of working with classifiers of the form $\mathrm{sgn}\circ g$. Because $\mathrm{sgn}$ is discontinuous, the Lipschitz-based analysis of Bubeck-Sellke does not directly extend to our setting. The co-stability term $S^{\ast}(g)$ compensates for this discontinuity: it quantifies how the score function interacts with the non-Lipschitz $\mathrm{sgn}$ map and is therefore crucial for establishing our classification-specific law of robustness.
> > >
> > > **(2):**
> > > First, we would like to distinguish the finite and infinite function class settings. In the _finite_ case Bubeck-Sellke derive their law of robustness for classes of _input-space Lipschitz_ functions, whereas in our setting we do not assume any Lipschitzness or other structural regularity on the function class in Theorem 4.1, and only very mild topological assumptions in Theorem 4.2. This is the sense in which our approach differs fundamentally from theirs in the finite-class case. Bubeck-Sellke do not explicitly state their finite-function-class result, although it follows directly from their Theorem 2. Similarly, using Theorem 4.1. we would obtain a looser bound (as previously discussed) in Corollary 6 which we do not explicitly present in the paper.
> > >
> > > To extend the analysis to _infinite_ function classes, a $\varepsilon$-discretization of the function class is necessary, leading to the Lipschitzness requirement in parameter space.
> > > In this respect, the technical requirement is indeed similar: Bubeck-Sellke assume that the function class is Lipschitz both in input space and parameter space, whereas in our case the Lipschitzness is imposed only on the score function $g$. The resulting classifier $\mathrm{sgn}\circ g$ remains discontinuous in both input and parameter space.
> > > Therefore, the term “unrelated” in our previous statement may be unintentionally confusing. It refers to the fact that one can obtain a law of robustness for classifiers without any structural assumption in input space (in contrast to Bubeck-Sellke) and the Lipschitzness in the parameter space is only assumed for a covering-number argument (as in Bubeck-Sellke).
> > >
> > > We are grateful for the reviewer’s careful consideration of our work and welcome further discussion on this or any additional points.

---

> > > > ### Comment · Reviewer_WFc2 · 2025-11-27
> > > >
> > > > Thank you for your reply. I still disagree with your claim that **if $S^{\*}(g)$ is small, the bound in Equation 8 becomes uninformative, and we cannot conclude anything meaningful about $L(g)$.**
> > > >
> > > > As I mentioned in my previous response, if $S^{\*}(g)$ is constant, no matter how small, I believe this would not have a significant impact from the perspective of overparameterization. For your claim "uninformative" to be sound, I think $S^{\*}(g)$ should depend on $n$, $p$, or $d$.

---

> > > > > ### Author Response · Authors · 2025-12-02
> > > > >
> > > > > We address the reviewer's final comment below.
> > > > >
> > > > > Note that $S^{\ast}(g)$ is not a fixed constant. Indeed, as the notation indicates, $S^{\ast}(g)$ depends on the function $g$, more precisely,
> > > > > $$
> > > > > S^{\ast}(g) = \mathbb{E}_{X}\left[ |g(X)| \right],
> > > > > $$
> > > > > for a function $g : \mathbb{R}^{d}\times \mathbb{R}^{p} \to [-1,1]$. In particular, $S^{*}(g)$ depends on the data distribution, the number of parameters, and the input dimension; it is not a constant independent of $p$ and $d$.
> > > > >
> > > > > Let us describe how this makes it necessary to consider the ratio $\frac{S^{\ast}(g)}{L(g)}$ in the overparameterization argument. Using our bound (Corollary 15), one obtains that if $\sqrt{d} \gg C$ and $p=n$, then
> > > > > $$
> > > > > \frac{S^{\ast}(g)}{L(g)} < \frac{C}{\sqrt{d}} \ll 1,
> > > > > $$
> > > > > showing that a classifier cannot be simultaneously stable (i.e. high normalized co-stability) and non-overparameterized.
> > > > >
> > > > > If one attempted instead to obtain a direct lower bound $L(g) \gg 1$, one would need to require
> > > > > $$
> > > > > S^{\ast}(g)\sqrt{d} \gg C \qquad \text{for all } g\in \mathcal{G}.
> > > > > $$
> > > > > However, the hypothesis class $\mathcal{G}$ may contain functions with $S^{\ast}(g)$ arbitrarily close to zero. A condition of the form $S^{\ast}(g)\sqrt{d} \gg C$ therefore cannot hold uniformly, since $S^{\ast}(g)$ itself depends on $d$ and cannot be bounded below by a universal constant. This is exactly what we mean when we say the bound becomes “uninformative’’: without controlling $S^{\ast}(g)$, Corollary 15 does not yield any meaningful statement about $L(g)$. We also note that deriving a nontrivial lower bound on $S^{\ast}(g)$ explicitly depending on $p,d$ is just as difficult as obtaining a tight upper bound on the Lipschitz constant.
> > > > >
> > > > > Finally, we emphasize that $S^{\ast}(g)$ and $L(g)$ play symmetric roles in the analysis. If one restricts the hypothesis class to functions with uniformly bounded Lipschitz constant $L(g)\le\mathrm{const}$, then our result yields information only about $S^{*}(g)$.
> > > > >
> > > > > We are happy to add a clarification to the paper that $S^{\ast}(g)$ is not a constant and implicitly dependent on p and d.

---

### Official Review · Reviewer_mVwX · 2025-11-03

**Soundness:** 4
**Presentation:** 4
**Contribution:** 3
**Rating:** 6
**Confidence:** 2

**Summary:**

The paper extends results on isoperimetry and robustness to discontinuous classifiers, instead of the regression setup from Bubeck & Sellke.
To this end the authors define a notion of class stability that measures the expected distance to the decision boundary within each class, and serves as a type of stability criterion in the classification case.
For a finite hypothesis class, a bound is derived for the Rademacher complexity that depends on the minimum class stability, the isoperimetry of the data distribution and of course the sizes of the class and dataset. The main conclusion from the bound is that the size of the hypothesis class needs to grow much larger than the size of the dataset in order to produce a good upper bound on the generalization error.

The results are extended to non-finite hypothesis classes using some additional conditions, and simulations are performed on MNIST and CIFAR10 to demonstrate the theory.

**Strengths:**

The paper provides a nice addition to the literature on stability and interpolation. While it gives results that are in similar flavor to existing results, there are still original developments that can be of interest to the community.
To achieve their generalization, the authors discuss several notions of stability and how they are combined with assumptions on the hypothesis class, in order to obtain meaningful results. I think these derivations are easy-to-understand and clearly written, and so is the rest of the paper.

**Weaknesses:**

The most apparent weakness of the paper is that it is somewhat incremental, and proves results that are in the spirit of Bubeck and Sellke. Since the technical tools developed in the paper are novel, I think that this is not a major drawback.

Small comment: the authors mention generalization also in the context of out-of-distribution generalization and refer to a few results on this problem (e.g. Zou et al. 24). Since the paper refers to these topics and mentions generalization as a whole and not just in-distribution (or adversarially robust) generalization, I think it should also make a distinction between this setting and the setting of robustness to other distribution shifts (like spurious correlations, covariate shift etc.). The settings are rather different, but there were works discussing overparameterization in the context of distribution shifts, giving both negative and positive results [1,2,3]. Since the two lines of work share some terminology, I think it'd be useful to shortly clarify the distinction.

[1] Wald, Y., Yona, G., Shalit, U., & Carmon, Y. Malign Overfitting: Interpolation Can Provably Preclude Invariance. In The Eleventh International Conference on Learning Representations.

[2] Hao, Y., Lin, Y., Zou, D., & Zhang, T. (2024). On the benefits of over-parameterization for out-of-distribution generalization. arXiv preprint arXiv:2403.17592.

[3] Sagawa, S., Raghunathan, A., Koh, P. W., & Liang, P. (2020, November). An investigation of why overparameterization exacerbates spurious correlations. In International Conference on Machine Learning (pp. 8346-8356). PMLR.

**Questions:**

No immediate questions come to mind, as the paper is written quite clearly. I will read the other reviews and see if questions come up from them.

---

> ### Author Response · Authors · 2025-11-21
>
> We thank the reviewer for their helpful feedback. In response to the comment on out-of-distribution (OOD) generalization, we have updated the paper by adding the following paragraph to the Related Work section:
>
> **Out-of-Distribution Generalization**: Classically and also in our analysis, generalization is based on the independently and identically distributed assumption on the data, in particular, the test data are generated from the same distribution as the training data coined In-Distribution (ID) generalization. In contrast, Out-of-Distribution (OOD) generalization aims to study the generalization performance under distributional shifts. To make the problem tractable the potential shifts are constrained to, for instance, spurious correlations or covariate shifts. In the OOD setting the connection between overparameterization and generalization has been studied in a series of works with positive [2] and negative [1,3] results.
>
> Adversarial robustness can be viewed as a special case of OOD generalization, where the distributional shift is constrained to lie within a perturbation set [4]. In this sense, our stability-based analysis is conceptually connected to OOD generalization. However, our results do not provide explicit bounds on OOD error; instead, we focus on ID generalization under the assumption that the classifier satisfies a given level of adversarial robustness expressed as the (co-)margin.
>
> [4] Aman Sinha, Hongseok Namkoong, Riccardo Volpi, John Duchi. Certifying Some Distributional Robustness with Principled Adversarial Training.  arXiv:1710.10571

---

> > ### Comment · Reviewer_mVwX · 2025-11-21
> >
> > Thank you for the response, I will keep my score and recommend acceptance.

---

### Author Response · Authors · 2025-12-03
**Summary of the rebuttal**

We thank all reviewers for their engagement and for the constructive discussion, which has helped improve the clarity and presentation of our work. We are pleased that we were able to resolve most concerns: Reviewer qsC1 kept their score, Reviewer mVwX explicitly recommends acceptance, and Reviewer WFc2 expressed satisfaction with our responses. Although the final exchange with Reviewer WFc2 remained open-ended, we are confident that our follow-up reply fully addresses the remaining point.

Below we summarize the main points of our rebuttal.

1. We added new experiments that further support our theoretical findings.
2. We empirically tested the isoperimetry assumption on Gaussian data,   MNIST, and CIFAR-10.
3. We expanded and improved the Related Work section.
4. We clarified several technical aspects raised by the reviewers.

More specifically:

1. We demonstrated a clear correlation between stability and test accuracy across a range of standard architectures – including MLPs, CNNs, and Heaviside networks. (See responses to qsC1 and WFc2.)


2. We introduced a structured empirical probe of isoperimetry and concentration. On Gaussian toy data, our estimator recovers the correct isoperimetric scale for linear functions and 1-Lipschitz networks; random MLPs behave as worst-case functions with very large effective constants; trained networks fall between these extremes. Applying the same procedure to MNIST and CIFAR-10, we observe near-isoperimetric behavior with moderate constants for the practically relevant function classes (linear probes, structured Lipschitz networks, trained margins), whereas worst-case Lipschitz models require far larger constants. This provides empirical support for the practical reasonableness of our assumptions in the regimes where the theory applies. (See response to qsC1.)

3. We explicitly clarified how our setting differs from general OOD generalization, and we integrated a concise discussion of OOD generalization into the Related Work section. (See response to mVwX.)

4. We clarified the relationship to Bubeck–Sellke and the nature of our contribution. Conceptually, both works link robustness and overparameterization, but the technical setting is different: Bubeck–Sellke work with continuous, globally Lipschitz regressors, whereas we handle genuinely discontinuous classifiers via two weaker, classification-specific notions of regularity (class stability and co-stability). It is not obvious a priori that such weaker, margin-based quantities still yield a law of robustness with the same scaling, and our main theoretical contribution is to show that they do.

   For finite hypothesis classes, our result requires no Lipschitz assumptions at all (and only mild topological ones in the sharpened version), unlike Bubeck–Sellke’s framework. For infinite classes, both works rely on Lipschitzness in the parameters purely as a standard tool to allow a  $\varepsilon$-discretization of the function class; the key difference is that we do not impose Lipschitzness on the classifier, which remains discontinuous as $\mathrm{sgn}\circ g$.


      We also clarified the role of co-stability $S^{\ast}(g)$: it is not a fixed constant but implicitly depends on the data distribution, model size, and the data dimensionality. The overparameterization result necessarily involves the ratio $\frac{S^{\ast}(g)}{L(g)}$. (See responses to WFc2.)

     We further explained that our structural assumptions cover essentially all standard deep-learning architectures (including MLPs, CNNs,
     RNNs, Transformers and more generally multiclass softmax models). Random forests and Heaviside networks, fall outside this structural   class but remain covered by our finite-class result. (See responses to WFc2 and qsC1.)

---

### Meta-Review · Area_Chair_fgKW · 2026-01-06

**Summary:**

This paper studies the relationship between overparameterization, robustness, and generalization for discontinuous classifiers. It introduces class stability and normalized co-stability as robustness notions, derives generalization bounds under isoperimetry, and shows that achieving both interpolation and robustness requires substantial overparameterization. The theory extends the law of robustness beyond Lipschitz predictors and is supported by experiments on MNIST and CIFAR-10.

Reviewers found the work technically solid, clearly written, and well-positioned relative to prior robustness and margin-based generalization theory, highlighting the novelty of handling discontinuous classifiers. Concerns focused on assumptions (e.g., isoperimetry), scope of experiments, and interpretation of constants.

The authors’ responses carefully addressed these points, clarified assumptions, strengthened empirical evidence, and resolved technical questions. Overall, the contributions are significant and well-justified, leading to acceptance.

**Reviewer Concerns:**

Please see my summary.

**Reviewer Scores:**

It is difficult to say.  Overall, the authors provided some solid rebuttal, but it's a subjective judgement for the reviewer whether they would like to raise their score.

---

### Decision · Program_Chairs · 2026-01-26

Accept (Poster)